# Self-assembling human heart organoids for the modeling of cardiac development and congenital heart disease

Yonatan R. Lewis-Israeli [1,2], Aaron H. Wasserman [1,2], Mitchell A. Gabalski[1,2], Brett D. Volmert[1,2], Yixuan Ming [3], Kristen A. Ball [1,2], Weiyang Yang [4,5], Jinyun Zou[3], Guangming Ni[3], Natalia Pajares[6], Xanthippi Chatzistavrou[6], Wen Li [4,5], Chao Zhou [3] & Aitor Aguirre [1,2✉]

Congenital heart defects constitute the most common human birth defect, however understanding of how these disorders originate is limited by our ability to model the human heart accurately in vitro. Here we report a method to generate developmentally relevant human heart organoids by self-assembly using human pluripotent stem cells. Our procedure is fully defined, efficient, reproducible, and compatible with high-content approaches. Organoids are generated through a three-step Wnt signaling modulation strategy using chemical inhibitors and growth factors. Heart organoids are comparable to age-matched human fetal cardiac tissues at the transcriptomic, structural, and cellular level. They develop sophisticated internal chambers with well-organized multi-lineage cardiac cell types, recapitulate heart field formation and atrioventricular specification, develop a complex vasculature, and exhibit robust functional activity. We also show that our organoid platform can recreate complex metabolic disorders associated with congenital heart defects, as demonstrated by an in vitro model of pregestational diabetes-induced congenital heart defects.

[1] Division of Developmental and Stem Cell Biology, Institute for Quantitative Health Science and Engineering, Michigan State University, East Lansing, MI, USA. [2] Department of Biomedical Engineering, College of Engineering, Michigan State University, East Lansing, MI, USA. [3] Department of Biomedical Engineering, Washington University in Saint Louis, St. Louis, MO, USA. [4] Division of Biomedical Devices, Institute for Quantitative Health Science and Engineering, Michigan State University, East Lansing, MI, USA. [5] Department of Electrical and Computer Engineering, College of Engineering, Michigan State University, East Lansing, MI, USA. [6] Department of Chemical Engineering and Material Science, College of Engineering, Michigan State University, East Lansing, MI, USA. ✉email: aaguirre@msu.edu

Cardiovascular disorders, including cardiovascular disease and congenital heart defects (CHD), are the leading cause of death in the developed world, and the most common type of congenital defect in humans, respectively. Despite the importance of understanding human cardiovascular disorders for treatment and prevention, progress on the creation of human heart organoid models for cardiovascular disease studies has been limited and lags significantly behind other organs (e.g., kidney, colon, intestine, brain)[1–4]. Human pluripotent stem cells (hPSCs) enable us to recapitulate important developmental steps in vitro to produce specific cardiac cell types with relative ease, high purity, and in large amounts[5–7]. However, current cell models are still far away from the structural and cellular complexity of the tissues and organs they intend to represent (e.g., lack of 3D matrix, disorganized cells, and absence of multicell-type interactions). These models frequently study isolated cell types and minimize or ignore other heart cells (e.g., epicardial cells, endocardial cells) or the contribution of cell-cell communication to a disease phenotype. There is a strong demand to bridge this technological and knowledge gap, as producing more faithful in vitro models of the human heart will allow us to better study healthy and diseased states for research and translational applications.

Significant attempts have been made over the last decade to address the lack of relevant heart-on-a-chip or heart organoid models, particularly using tissue engineering techniques[8–15]. While these approaches allow for high control of the end construct, they tend to be expensive, work-intensive, and not readily scalable. Furthermore, they frequently do not faithfully represent the original cell composition (e.g., use of dermal fibroblasts or HUVECs)[8,9] and organization (e.g., cardiospheres[10,11]) of the heart. These approaches yield functional tissues but fall short in terms of physiological and structural relevance, as well as cell and ECM complexity. In more recent times, self-assembling organoid technologies have become available for the heart. These approaches are exploring the differentiation of PSC embryo-like aggregates in an attempt to recapitulate early cardiogenesis in vitro. Recently, mouse embryonic stem cells (ESCs) were used to generate precardiac organoids showing distinct heart field specifications[16], cardiac crescent-like structures juxtaposed with primitive gut tube[17], and atrial and ventricular cardiomyocyte lineages[18]. These studies provided us with a great deal of understanding and information of early heart development in vitro, but are faced with limitations associated with mouse models. Hybrid cardiac-foregut organoids from human PSCs have also been reported very recently, with external heart layers and an internal endodermal core[19] and cardioids (an alternative nomenclature for heart or cardiac organoids) have been reported with a large internal chamber, relevant cardiac cell lineages and single cell transcriptomic analysis[20]. While demonstrating a large leap in self-assembling capabilities, these cardioids still relied heavily on complex growth factor mixtures, co-culture approaches to include epicardial clusters, and independent differentiation protocols to achieve their final results.

Here, we report a small molecule-based methodology to create highly complex and physiologically relevant self-assembling human heart organoids (hHOs) using hPSCs by manipulating cardiac developmental programs. Our protocol relies mainly on three sequential Wnt modulation steps (activation/inhibition/activation) at specific time points on suspension embryoid bodies (EBs), and produces significant heart-like structures in terms of structure, organization, functionality, cardiac cell type complexity, ECM composition, and vascularization. Our method is low-cost when compared to growth factor-based approaches and involves less steps and manipulation than currently existing methods. It is also automatable, scalable, and amenable to high-content/high-throughput pharmacological screenings. As a proof-of-concept of the value of this system to model human cardiac disease, we utilized our organoid system to model the effects of pregestational diabetes (PGD) — clinically defined as diabetes before pregnancy and present during at least the 1st trimester of fetal development — on the developing embryonic heart.

## Results

**Self-assembling human heart organoids generated by Wnt signaling modulation.** Our method is designed to meet four initial milestones: (1) high organoid quality and reproducibility; (2) high-throughput/high-content format; (3) relative simplicity (no need for special equipment outside of traditional cell culture instrumentation); (4) defined chemical conditions for maximum control and versatility for downstream applications. We started by assembling hPSCs into embryoid bodies (EBs) by centrifugation in ultra-low attachment 96-well plates followed by a 48-h incubation at 37 °C and 5% $CO_2$ prior to induction (Fig. 1a). After induction, two-thirds of the spent medium was removed and replaced with fresh medium at each medium change, resulting in gradual transitions in exposure to the different signals employed and minimizing agitation of the organoids at the bottom of the well. Induction of mesoderm and cardiogenic mesoderm was achieved by sequential exposure to CHIR99021, a canonical Wnt pathway activator (via specific GSK3 inhibition), and Wnt-C59, a Wnt pathway inhibitor (via PORCN inhibition) (Fig. 1a), in a modification of previously described protocols[14,21,22]. Brightfield and immunofluorescence imaging of hHOs showed a significant increase in size throughout the differentiation protocol (Fig. 1b). Confocal microscopy for the cardiomyocyte-specific marker TNNT2 showed that organoids developed sarcomeres as early as day 7 (Fig. 1b), with much clearer sarcomere banding and fiber assembly by day 15 (Fig. 1c). Beating hHOs appeared as early as day 6 of the differentiation protocol, with robust and regular beating by day 10 in all organoids and lasting for at least 8 weeks in culture (Supplementary Movies 1–3, corresponding to the longest time tested). We found out that optimal conditions for the first Wnt activation were critical for successful heart organoid formation and differed from those reported for cardiac lineage monolayer differentiation. We exposed EBs to different concentrations of CHIR99021 (1 μM, 2 μM, 4 μM, 6.6 μM, and 8 μM) on day 0 for 24 h and then evaluated hHOs for cardiac lineage formation by confocal microscopy at day 15 of differentiation (Supplementary Fig. 1a). Optimal cardiogenic mesoderm induction for all human embryonic stem cell (hESC) and induced pluripotent stem cell (hiPSC) lines was found at 1–4 μM CHIR99021 concentrations, rather than the typical 10–12 μM range reported for monolayer methods[14,21–27]. 4 μM CHIR99021 exposure resulted in the highest cardiomyocyte content with 64.9 ± 5.3% TNNT2+ cells at day 15 (Fig. 1d, Supplementary Fig. 1a). We believe enhanced differentiation at lower CHIR concentrations can be attributed to endogenous morphogen production due to the self-assembling conditions provided[28–32]. Time-course RNA sequencing analysis between days 0 and 19 of differentiation supports this hypothesis by revealing the stepwise production of fundamental cardiac development morphogens and growth factors and their respective receptors (Suppl. Fig. 1b). hHOs treated with 4 μM CHIR99021 also displayed the best functional properties out of all tested concentrations (Suppl. Fig. 1c, d). Our hHO differentiation protocol was reproducible across multiple hPSC lines (iPSC-L1, AICS-37-TNNI1-mEGFP, iPSCORE_16_3, H9), all of which exhibited similar differentiation efficiencies, beat metrics, growth rates, and sizes (Fig. 1e, f).

To increase organoid complexity and produce more developmentally relevant structures, we developed a method to induce proepicardial organ specification based on a second Wnt

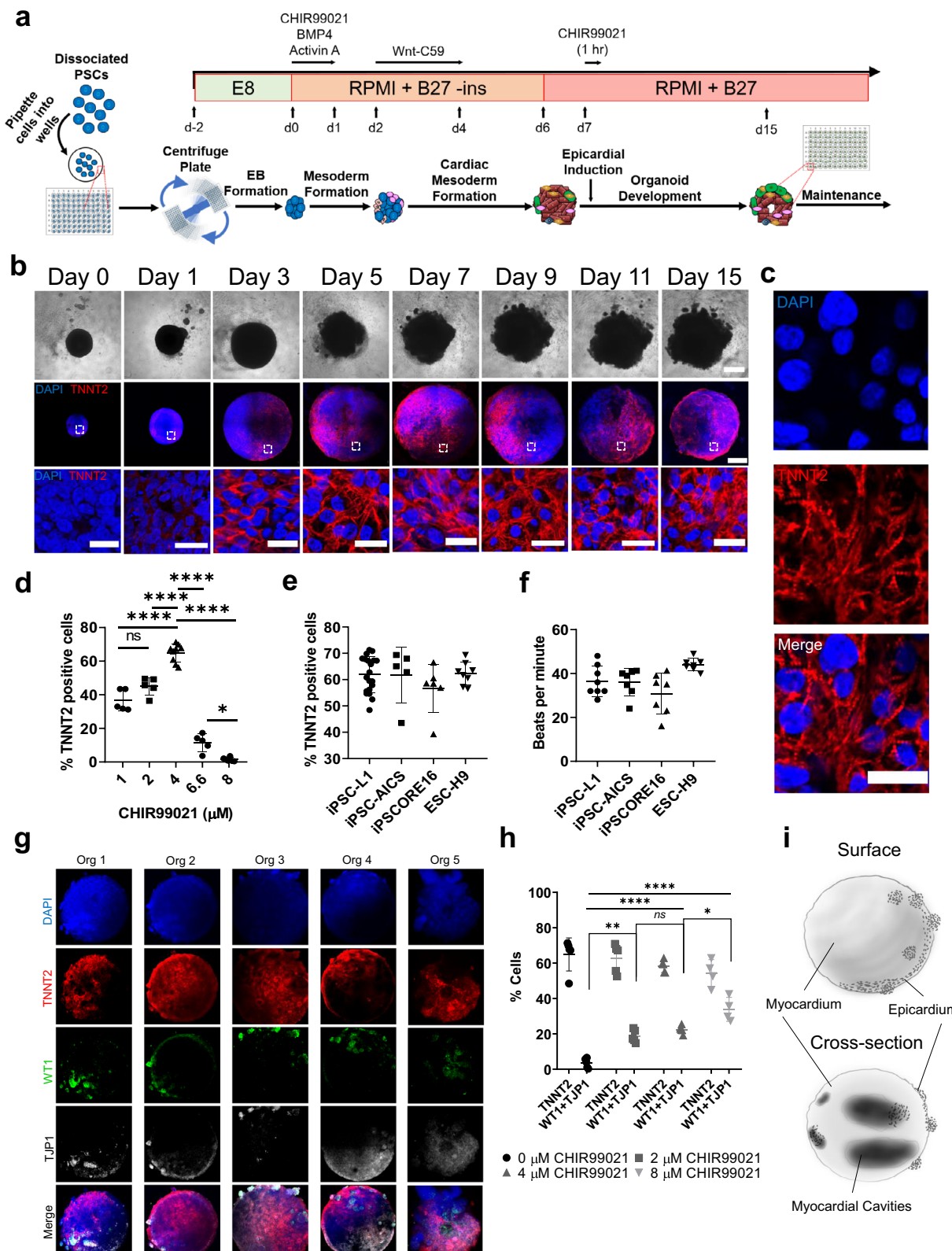

activation step[22] on differentiation days 7–9. To determine if this second activation would prime our hHOs to increase complexity and better recapitulate heart development, we tested the effects of a second CHIR99021 exposure on day 7 and continued culturing the hHOs to day 15 for fixation and imaging (Fig. 1a). Conditions for the second Wnt activation with CHIR99021 were determined by testing developing hHOs at varying concentrations (2, 4, 6,

and 8 μM), and exposure durations (1, 2, 12, 24, and 48 h). The efficiency of epicardial cell and cardiomyocyte formation was evaluated using confocal imaging and quantification for well-established epicardial (WT1, TJP1) and cardiomyocyte (TNNT2) markers at day 15 (Fig. 1g, h; Supplementary Fig. 2a–c). We found that the treatment robustly promoted the formation of proepicardium and epicardial cells (Fig. 1g; Supplementary

**Fig. 1 Three-step WNT signaling modulation triggers heart organoid formation in 3D culture. a** A schematic diagram depicting the protocol used to differentiate TNNT2 + cardiomyocytes in embryoid bodies. CHIR99021 concentration is variable at day 0 and day 7. (Blue cells represent PSCs, red cells represent cardiomyocytes, all other colors represent non-myocyte cells). **b** Brightfield images of developing organoid over 15 days of differentiation (top), confocal immunofluorescence images for DAPI (blue), and TNNT2 (red), of representative organoids by day from day 0 to 15; scale bar: 500 μm, inset: 50 μm (*n* = 20). **c** Confocal immunofluorescence images for DAPI (blue) and TNNT2 (red), in day 15 organoids differentiated using 4 μM CHIR99021 showing sarcomere bands; Scale bar: 25 μm. **d** Cell quantification of cardiomyocytes within organoids taken at multiple z-planes as a percentage of TNNT2 + cells to total cells of each organoid for the five CHIR99021 concentrations (*n* = 5 organoids). Value = mean ± s.d., 1 way ANOVA multiple comparison test; *$*p = 0.02$, ****$p < 0.0001$, ns: no significance $p = 0.08$. **e** Percentage of TNNT2 + cells normalized to total cells in confocal images of hHOs (*n* = 20 organoids), and **f** beating frequency (*n* = 7 organoids), in 3 iPSC lines and 1 ESC line. Value = mean ± s.d. **g** Confocal immunofluorescence images of hHOs at differentiation day 15 for DAPI (blue), WT1 (green), TNNT2 (red), and TJP1 (white) after day 7 epicardial induction with 2 μM CHIR99021 (*n* = 12 organoids). **h** Cell quantification of cardiomyocyte (TNNT2 +) and epicardial cells (WT1 + and TJP1 +) within organoids taken at multiple z-planes as a percentage of total cells of each day 15 organoid (*n* = 5 organoids). Value = mean ± s.d., 1 way ANOVA multiple comparison test; *$*p = 0.04$, **$**p = 0.0023$, ****$p < 0.0001$, otherwise ns: no significance $p = 0.9$. E8: Essential 8 media, RPMI + B27 – ins: RPMI with B27 supplements without insulin. **i** Sketch of hHOs showing surface (top) and cross-section (bottom) features of the hHOs. Source data are provided as a Source Data file.

---

Fig. 2a, b) on the organoid's surface, however, high concentrations or long exposure times inhibited cardiac cell type formation and provoked an undesired extensive epicardial expansion (Supplementary Fig. 2a, c, d). A single 2 μM CHIR99021 exposure for 1 h on differentiation day 7 produced the most physiologically relevant epicardial to myocardial ratio (60–65% cardiomyocytes, 10–20% epicardial cells) (Fig. 1g, h; Supplementary Fig. 2). Structurally, a significant part of the epicardial tissue was found on external layers of the organoid and adjacent to well-defined myocardial tissue (TNNT2 +) (Fig. 1g, Supplementary Fig. 2a, b), thus recapitulating the structural organization found in the heart. The robust expression of TJP1 on epicardial cell membranes also confirmed the epithelial phenotype of these cells (Supplementary Fig. 2b). Overall, the resulting hHOs contained significant myocardial tissue with epicardial tissue clustered near the outer surface of the organoids, mimicking the anatomical structure of the developing embryonic heart (Fig. 1i).

**Transcriptomic analysis reveals hHOs closely model human fetal cardiac development and produce all main cardiac cell lineages**. To characterize the developmental steps and molecular identity of the cellular populations present in hHOs, we performed transcriptomic analysis throughout hHO formation. hHOs were collected for RNA-sequencing at different time points (day 0 through day 19) of differentiation (Fig. 2, Supplementary Fig. 3). Unsupervised K-means clustering analysis revealed organoids progressed through three main developmental stages: day 0–day 1, associated with pluripotency and early mesoderm commitment; day 3–day 7, associated with early cardiac development; and day 9–day 19, associated with fetal heart maturation (Fig. 2a, Supplementary Fig. 3). Gene ontology biological process analysis identified important genetic circuitry driving cardiovascular development and heart formation (Fig. 2a and Supplementary Dataset 1; raw data deposited in GEO under "GSE153185"). To compare cardiac development in hHOs to that of previously existing methods, we performed RNA-seq on monolayer iPSC-derived cardiac differentiating cells using previously established protocols[3]. We also compared our RNA-seq results to publicly available datasets from previously reported monolayer cardiac differentiation protocols and human fetal heart tissue (gestational age days 57–67)[33] ("GSE106690"). In all instances, hHO cardiac development transcription factor expression regulating first and second heart field specification (FHF, SHF, respectively) was similar to that observed in monolayer PSC-derived cardiac differentiation and corresponded well to that observed in fetal heart tissue (Fig. 2b, Supplementary Fig. 3a). Gene expression profiles showed hHOs had higher cardiac cell lineage complexity than cells that underwent monolayer differentiation, especially in the

epicardial, endothelial, endocardial, and cardiac fibroblast populations (Fig. 2c, Supplementary Fig. 3b, c). These data suggest a significant enrichment in the structural and cellular complexity of our hHOs, thus bringing them in line with fetal hearts and further away from monolayer-based differentiation. This was confirmed by extending our gene expression analysis to look at several widespread critical gene clusters involved in classic cardiac function, including conductance, contractile function, calcium handling, and cardiac metabolism, among others (Fig. 2d). Of special interest, expression of heart-specific extracellular matrix genes was high in hHOs and fetal hearts but completely absent in monolayer differentiation protocols (Fig. 2d, Supplementary Fig. 2d). Markers for pluripotency were not found in hHOs beyond day 1 (Supplementary Fig. 3e). Principal component analysis showed a clear progression in development in the hHOs from day 0 to 19 (Supplementary Fig. 3f). Taken together, these data suggest hHO expression profiles are similar to those of fetal hearts, and their global transcriptomes are closer to those of fetal hearts than monolayer protocols, as determined by hierarchical clustering (Fig. 2e).

**Heart organoids produce multiple cardiac-specific cell lineages**. Results from the transcriptomic analysis (Fig. 2) suggested that the second CHIR99021 exposure led to the formation of other mesenchymal lineages and higher complexity in hHOs due to induction of proepicardial organ formation. To evaluate this finding, we performed immunofluorescence analysis for secondary cardiac cell lineages. Confocal imaging confirmed the presence of cardiac fibroblasts positive for THY1 and VIM (Fig. 3a), similar to the composition of the fetal heart[34]. Immunofluorescence analysis for the endocardial marker NFATC1 (an endocardial specific cell marker[19]) revealed the formation of endocardial layers, similar to the endocardial lining of heart chambers (Fig. 3b). Further imaging revealed a robust interconnected network of endothelial cells (PECAM1 +), and vessel-like tube formation throughout the organoid assembling between day 11 and 13 (Fig. 3c, d, Supplementary Fig. 4a). Higher magnification images uncovered a complex web of endothelial cells adjacent to or embedded in myocardial tissue (Fig. 3e, Supplementary Movies 4, 5). These results strongly indicate that during hHO development, endothelial vascular structures emerge, adding a vascular network to the organoids. Figure 3f shows a quantification of the contribution of the different cardiac cell populations to the organoids, with a composition of 12.49 ± 1.01% cardiac fibroblasts, 13.82 ± 1.54% endocardial cells, and 1.63 ± 0.21% endothelial cells. It should be noted that these non-myocyte cardiac cells were often intermixed within TNNT2 + myocyte regions (Fig. 3a–e) as seen in vivo[35]. Lastly,

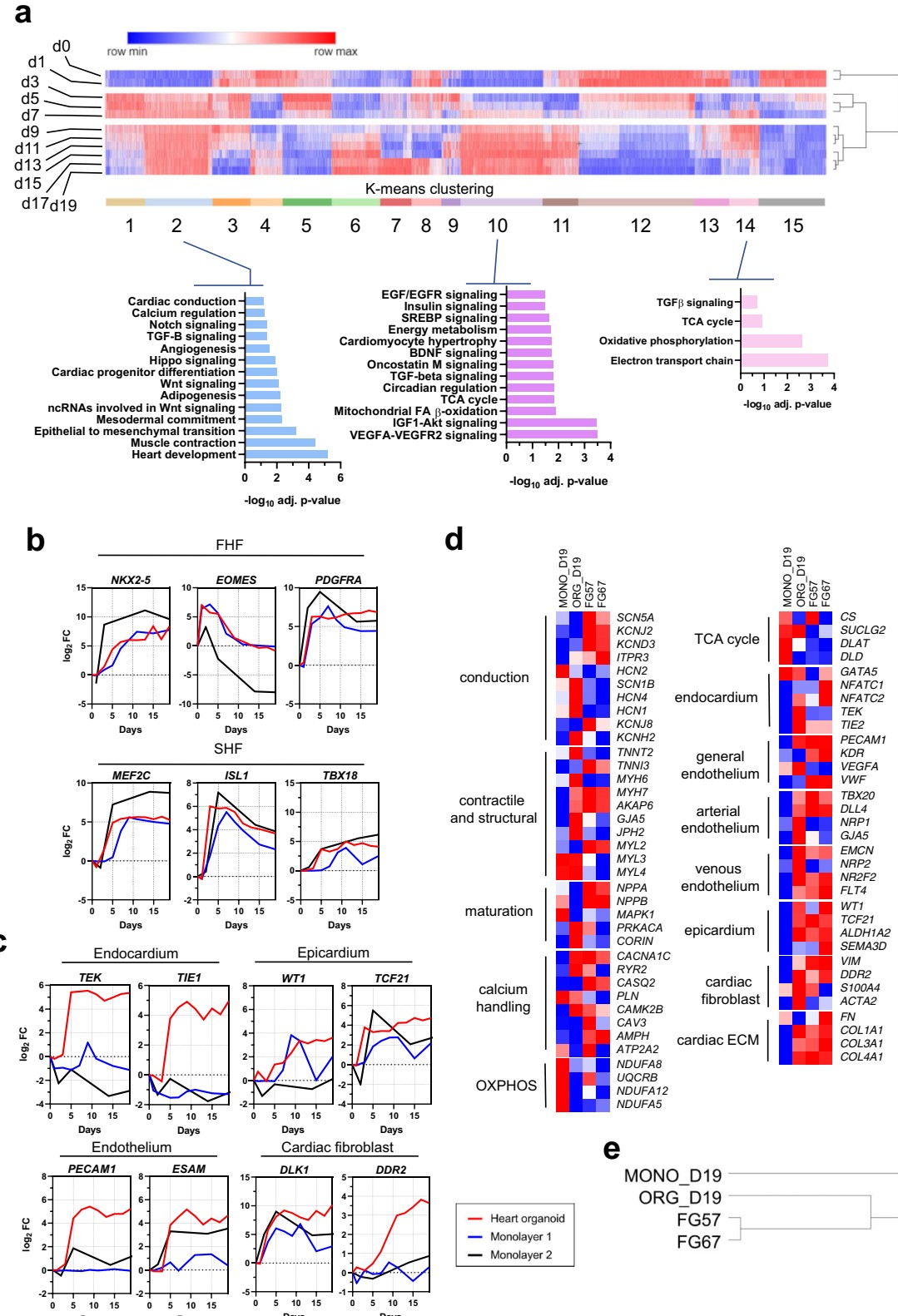

extracellular matrix proteins common in cardiac tissue, such as collagen type 1, collagen type 4, and fibronectin, were all observed in our hHO model (Supplementary Fig. 4b–d). Together, these observations depict a complex and sophisticated human heart organoid model with endocardial lined chambers within myocardial tissue, interspersed with cardiac fibroblasts and a network of endothelial cells, and complete with external epicardial tissue,

features that are highly recapitulative of the developing human heart (Fig. 3g).

**Human heart organoids recapitulate heart field development and atrioventricular specification**. The first and second heart fields (FHF, SHF) are two cell populations found in the developing heart. Cells from the FHF contribute to the linear heart

**Fig. 2 Heart organoids contain multi-cell type complexity and possess developmental and maturation characteristics similar to embryonic fetal hearts.**
**a** K-means cluster analysis of heart organoid transcriptomes by RNA-seq. Clusters strongly associated with fetal heart development (e.g. 2, 10, and 14) appear from day 9 onwards. Pathway enrichment analysis is provided below for representative cardiac-specific clusters. **b** Gene expression analysis ($\log_2$ fold-change vs. day 0) of first and second heart field markers over heart organoid differentiation process (FHF, SHF respectively). **c** Gene expression analysis ($\log_2$ fold-change vs. day 0) for cardiac-specific cell type populations in heart organoids, including epicardial cells, fibroblasts, endocardial cells, and endothelium. **d** Normalized comparison of key genes involved in cardiac function across heart organoids, monolayer differentiation methods, and fetal hearts at gestational day 57–67[33]. **e** Hierarchical clustering analysis of heart organoids, monolayer differentiation, and fetal hearts. FG57/67: fetal gestation day 57/67, FHF: first heart field, MONO_D19: monolayer cardiomyocytes at day 19 of differentiation, ORG_D19: human heart organoids at day 19 of differentiation, OXPHOS: oxidative phosphorylation, SHF: second heart field. Heatmap colors are relative intensity representing gene expression.

tube formation, followed by migrating cells belonging to the SHF that contribute to further expansion and chamber formation[36]. We found evidence of cells representing both heart fields in our organoids. HAND1 (FHF) and HAND2 (SHF) are members of the Twist family of basic helix-loop-helix (bHLH) transcription factors that play key roles in the developing heart[37]. Immunofluorescence of day 7 hHO cryosections showed well-differentiated regions of HAND1 and HAND2 (Fig. 4a) cells in the same organoid, suggesting that both FHF and SHF progenitors are present and segregated into their respective heart fields. FHF markers were observed as early as day 3 (prior to TNNT2 detection), confined to specific regions on the organoid, and reducing in expression around day 9 (Fig. 4b, Supplementary Fig. 5a). SHF markers appeared later in the process, only becoming prominent around day 7, and expressing throughout the organoid up to day 9 (Fig. 4b, Supplementary Fig. 5b). In human hearts, the left ventricle ultimately forms from FHF progenitors, and the atria form from SHF progenitors[38]. Therefore, we sought to determine if our hHOs contain cardiomyocytes committed to either the atrial or ventricular lineages. Immunofluorescence for MYL2 (encodes myosin light chain-2, ventricular subtype) and MYL7 (encodes myosin light-chain 7, atrial subtype) in day 15 hHOs showed cardiomyocytes positive for both subtypes. The two different populations localized to different regions of the organoid and were in close proximity, which mirrors the expression pattern seen in human hearts (Fig. 4c, d). Atrial cardiomyocytes made up most of the cell population (~48%) while ventricular cardiomyocytes made up about one fifth (~18%) of the total cells in the organoids at day 15 (Fig. 4e). The expression of HAND1, HAND2, and MYL7 transcripts was also observed throughout the differentiation protocol by RNA-seq and was similar to that observed in human fetal hearts (Supplementary Fig. 3a, c). We added a contrast dye (India ink) to organoid medium to reveal structural detail and record beating organoids under a light microscope. This revealed the presence of large central chamber-like structures surrounded by beating tissue, as suggested before by confocal imaging (Supplementary Movie 6). Taken together, these data suggest that the differentiation of our hHOs involves heart field formation, atrioventricular specification and chamber formation, all of which further emphasizes their recapitulation of human cardiac development.

**BMP4 and Activin A improve heart organoid chamber formation and vascularization**. The growth factors bone morphogenetic protein 4 (BMP4) and Activin A have frequently been used as alternatives to small molecule-mediated Wnt signaling manipulation, since they are the endogenous morphogens that pattern the early embryonic cardiogenic mesoderm and determine heart field specification in vivo[16,39–41]. We suspected that BMP4 and Activin A, in combination with our small molecule CHIR activation/inhibition protocol, could synergistically improve the ability of hHOs to recapitulate cardiac development in vitro. We tested the effect of BMP4 and Activin A in the context of our optimized protocol by adding the two morphogens

at 1.25 ng/ml and 1 ng/ml, respectively[16], at differentiation day 0 for 24 h in conjunction with 4 μM CHIR99021. No significant differences were found in the formation of myocardial (TNNT2+) or epicardial (WT1+/TJP1+) tissue between control and treated hHOs (Fig. 5a). However, significant differences in organoid size were observed as hHOs treated with growth factors were about 15% larger in diameter (Fig. 5b, c). This difference may correspond with increased chamber connectivity, as BMP4/Activin A-treated hHOs had internal chamber-like cavities that were ~50% more interconnected with other chambers compared to control hHOs (Fig. 5d, e). Notably, immunofluorescence analysis of organoids treated with BMP4 and Activin A showed a 160% increase in PECAM1+ cells too, indicating a significant effect on organoid vascularization (Fig. 5f, g). These results suggest improved structural organization in the developing organoids under BMP4/ActA, in agreement with previously reported differentiation methods using BMP4 and Activin A alone[16].

**Heart organoids exhibit functional and structural features of the developing human heart**. Traditional imaging methods, such as confocal imaging, are poorly suited for the study of the complex 3D structures of the size present in hHOs. Thus, we employed optical coherence tomography (OCT) to characterize chamber properties using minimally invasive means, thereby preserving chamber physical and morphological properties. OCT showed clear chamber spaces within day 15 hHOs (Fig. 6a, Supplementary Fig. 6a–c). 3D reconstruction of the internal hHO topology revealed a high degree of interconnectivity between these chambers (Supplementary Movies 7–10), revealing 4–6 chambers near the center of the organoid ranging from $5.5e^{-4}$ mm$^3$ to $1.3e^{-2}$ mm$^3$ (Supplementary Fig. 6d, Supplementary Movie 10). Given the relatively large size of our heart organoids (up to 1 mm in diameter, ~0.45 mm$^3$), we decided to verify whether these chambers could be attributed to internal cell death. We created a transgenic hiPSC line expressing FlipGFP, a non-fluorescent engineered GFP variant that turns fluorescent upon effector caspase activation and is thus a reporter for apoptosis[42]. FlipGFP organoids in control conditions exhibited no fluorescence indicating that there is no significant programmed cell death (Supplementary Fig. 6e). Doxorubicin-treated hHOs were used as a positive control for apoptosis (Supplementary Fig. 6e), with evident signs of cell death.

Ultrastructural analysis of hHOs showed similar features to those found in early human fetal hearts, with well-defined sarcomeres surrounded by mitochondria, gap junctions and the presence of tubular structures reminiscent of T-tubules (Fig. 6b), also confirmed by immunofluorescence staining with WGA (Fig. 6c, Supplementary Fig. 6f). We also measured electrophysiological activity to determine the hHO functionality. Utilizing a multi-electrode array (MEA) (Supplementary Fig. 7), we could detect robust beating and normal electrophysiological activity with well-defined action potential waves reminiscent of QRS complexes, T and P waves, and regular action potentials across multiple organoids (Fig. 6d). We also performed live calcium imaging in whole organoids to determine calcium activity. We generated hHOs from an iPSC line expressing the

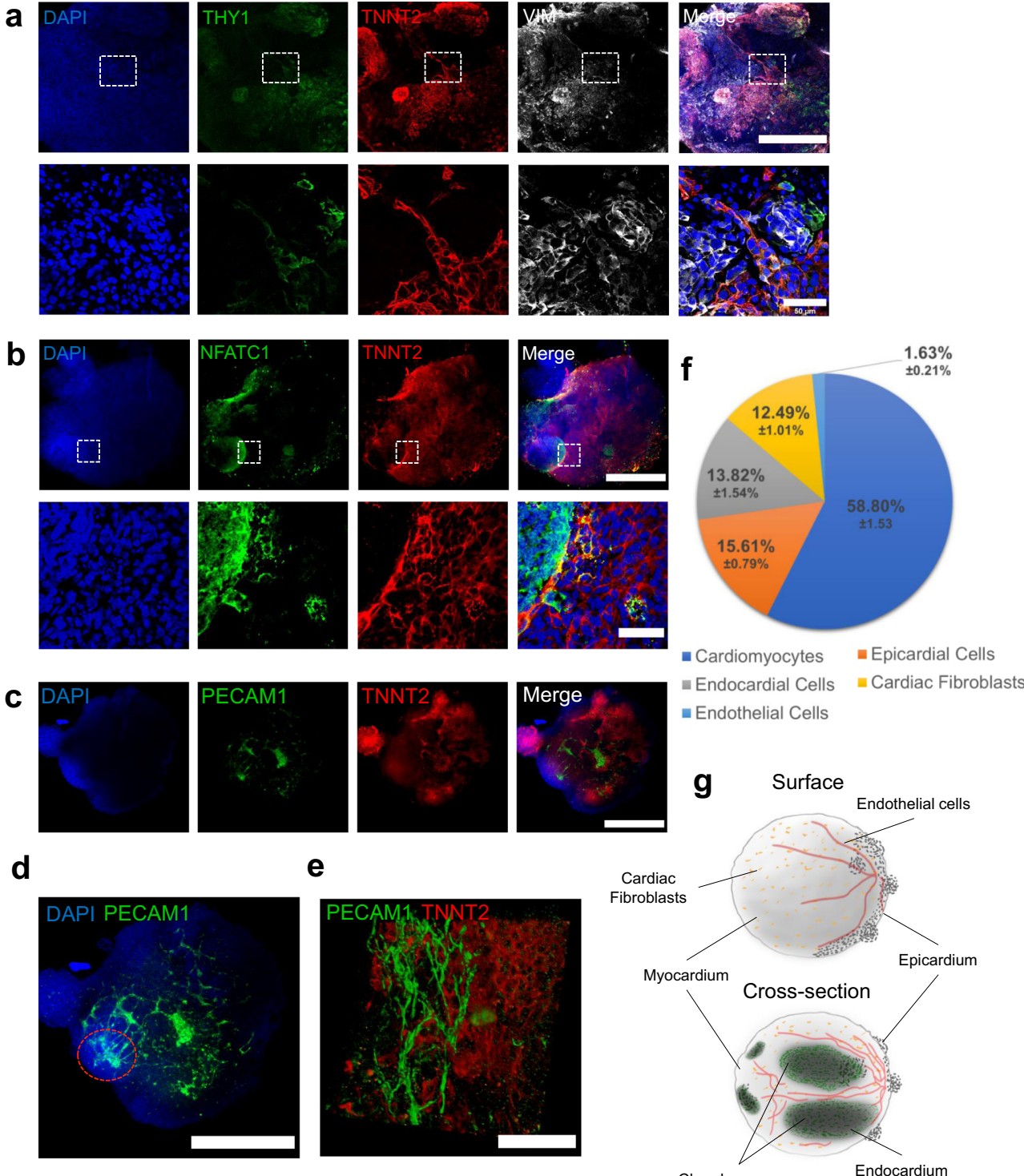

**Fig. 3 hHO cardiac cell lineage composition. a–e** Immunofluorescence images of various cell lineages composing the hHOs. **a**, Cardiac fibroblast markers THY1 (green) and VIMENTIN (white) present throughout the hHOs, TNNT2 (red), DAPI (blue); scale bar: 500 μm, inset: 50 μm. **b** Endocardial marker NFATC1 (green) highly expressed within chambers of TNNT2 $^+$ tissue (red); scale bar: 500 μm, inset: 50 μm. **c–e** Endothelial marker PECAM1 (green) showing a defined network of vessels throughout the organoid and adjacent to TNNT2 $^+$ tissue (red), DAPI (blue); showing a single confocal plane (**c**), a maximum intensity projection to visualize the vascular network throughout the organoid (**d**), and a high magnification 3D reconstruction showing tubular endothelial structures (**e**); red dotted circle in (**d**) indicates area of high vascular branching. Scale bar: **c**, **d**: 500 μm, **e** 50 μm. **f** Pie chart of average cell composition in hHOs, calculated as the percentage of cells with respective cell marker over all cells by nuclei counting using ImageJ. **g** Sketch of hHO surface (top) and cross-section (bottom) showing the organization of cell types and features of the hHOs. Source data are provided as a Source Data file.

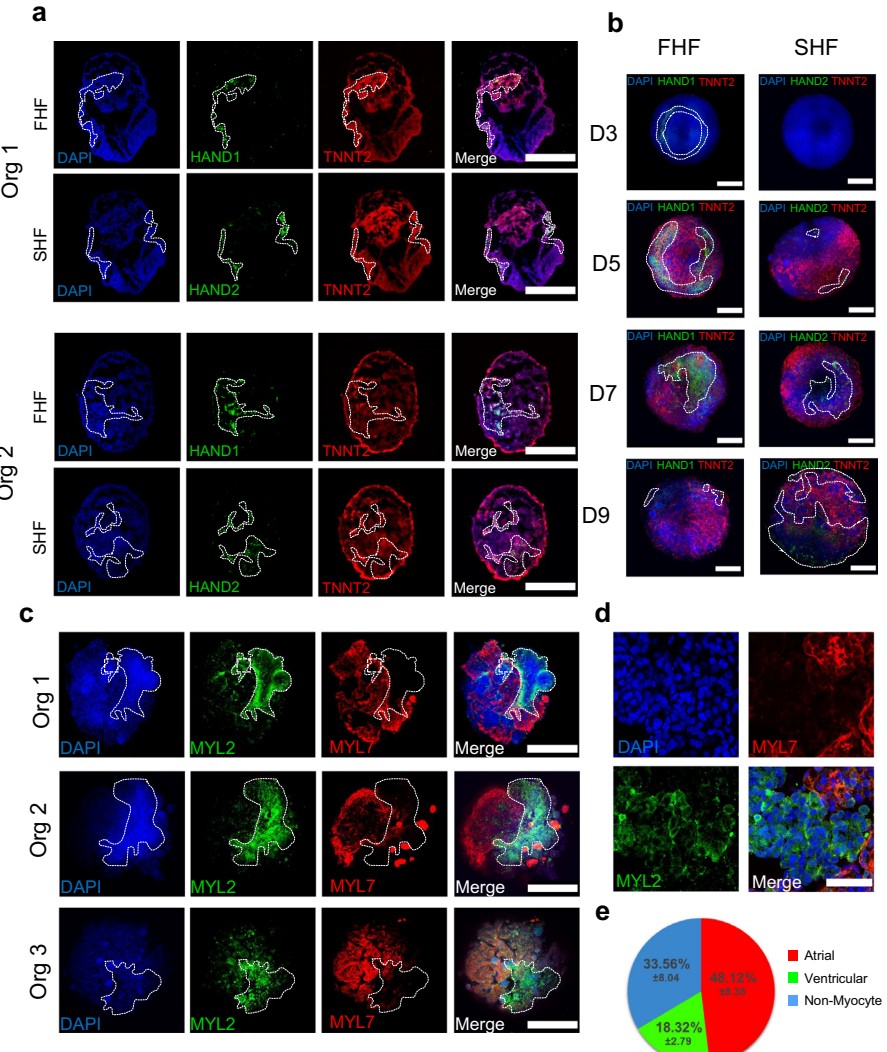

**Fig. 4 Heart field development and cardiomyocyte specification in human heart organoids. a** Confocal immunofluorescent images of two representative hHO cryosections on day 8 of differentiation showing robust HAND1 (FHF) and HAND2 (SHF) transcription factor expression (green) in different sections of the same organoid; TNNT2 (red), DAPI (blue); scale bar: 500 μm, (n = 12 organoids). **b** Day 3 to day 9 hHOs showing formation of FHF (HAND1, left) and SHF (HAND2, right); scale bar: 500 μm. **c** Confocal immunofluorescence images of three representative day 15 hHOs containing well-differentiated ventricular (MYL2, green) and atrial (MYL7, red) regions, DAPI (blue); scale bar: 500 μm (n = 10). **d** Inset images of organoid 1 from (**c**) scale bar: 50 μm. **e** Pie chart showing the percentage of atrial cardiomyocytes (MLC2a + DAPI), ventricular cardiomyocytes (MLC2v + DAPI), and non-myocyte cells (DAPI only); value = mean ± SD, (n = 6 organoids). FHF: first heart field, Org: organoid, SHF: second heart field. Source data are provided as a Source Data file.

rapid calcium indicator GCaMP6f[43,44], and imaged fluorescence variation over time as a result of calcium entry and exit from the cells. hHOs presented strong regular calcium waves typical of cardiac muscle and in agreement with our electrophysiology data (Fig. 6e, Supplementary Movie 11).

**Modeling pregestational diabetes induced CHD**. As proof-of-concept on the utility of our system, we used our hHO model to study the effects of pregestational diabetes (PGD) on cardiac development. Diabetes affects a large sector of the female population in reproductive age and comes associated with significant epidemiological evidence linking it to CHD during the first trimester of pregnancy (up to 12-fold risk increase, 12% vs. 1% risk for healthy females), but little understanding of the underlying mechanisms exists, especially in humans. To model this condition, we modified hHO culture conditions to reflect reported physiological levels of glucose and insulin in normal mothers (3.5 mM glucose, 170 pM

insulin, normoglycemic hHOs or NHOs)[45], and reported diabetic conditions for females with type I and type II pregestational diabetes (11.1 mM glucose and 1.14 nM insulin, pregestational diabetes hHOs or PGDHOs)[45,46]. NHOs and PGDHOs showed significant morphological differences as early as day 4 of differentiation. NHOs were slower to grow and exhibited patterning and elongation between days 4 and 8, while PGDHOs remained spherical throughout the two-week period (Fig. 7a; Supplementary Fig. 8a). PGDHOs were also significantly larger after 1 week of differentiation (Fig. 7b), suggesting macrosomia, a common outcome of newborns born to diabetic mothers[47]. Electrophysiological analysis showed irregular frequency of action potentials in PGDHOs suggesting arrhythmic events (Fig. 7c and Supplementary Fig. 8b, c). Metabolic assays for glycolysis and oxygen consumption revealed decreased oxygen consumption rate in PGDHOs and increased glycolysis when compared to cells dissociated from NHOs (Fig. 7d, e, Supplementary Fig. 8d). TEM imaging revealed PGDHOs had a reduced number of mitochondria surrounding sarcomeres and a significantly higher number of lipid

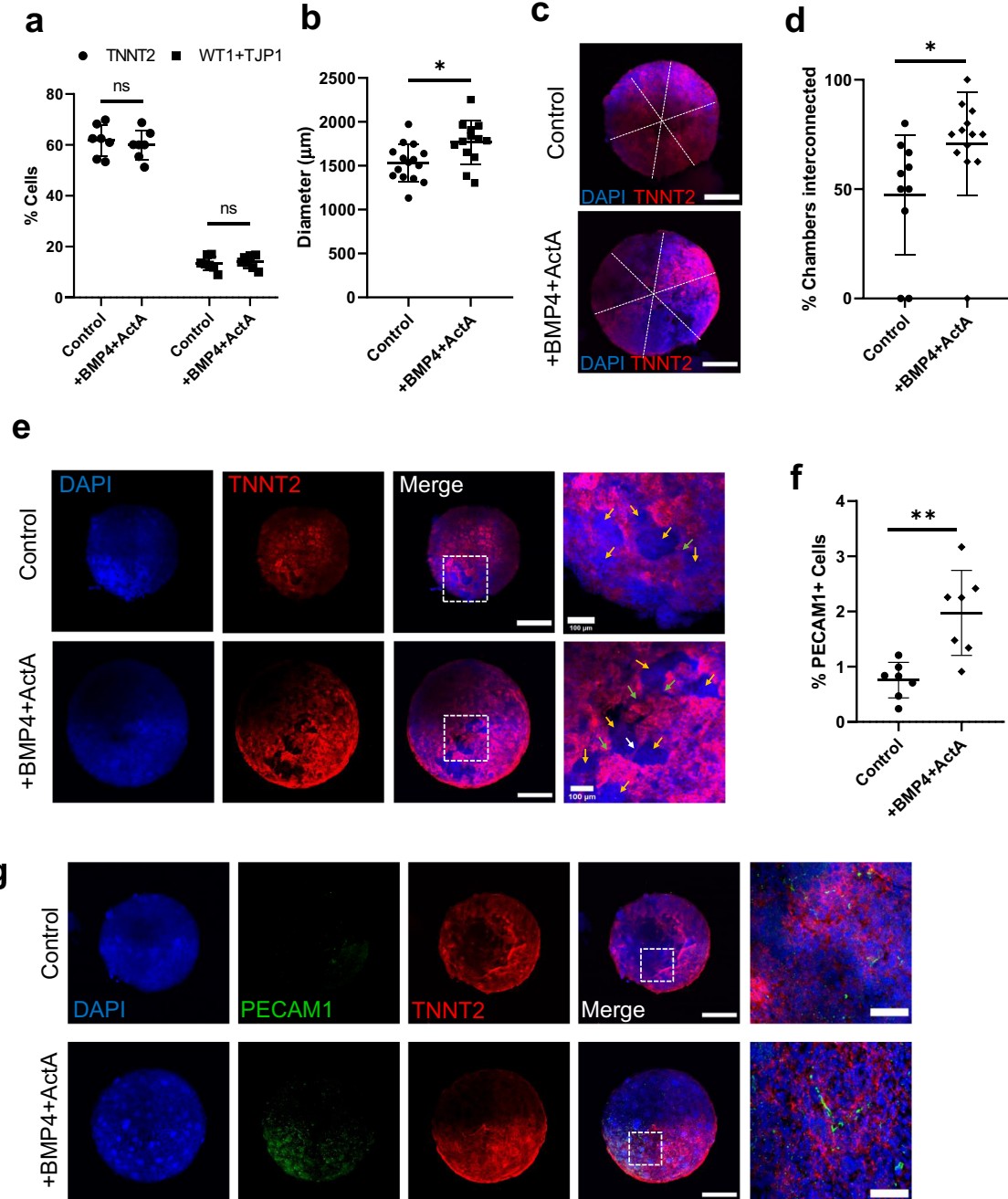

**Fig. 5 BMP4 and Activin A improve heart organoid differentiation and development. a–g** All panels compare hHOs differentiated with CHIR99021 alone (control) and with CHIR + BMP4 + Activin A (treated). **a** Percent of cardiomyocyte and epicardial positive cells as a percentage of total DAPI + nuclei ($n = 7$ organoids per condition, value = mean ± s.d., 2-way ANOVA Sidak's multiple comparisons test, ns: $p = 0.76$ for TNNT2 and $p = 0.97$ for WT1 + TJP1) and **b** organoid diameter, ($n = 13$ organoids per condition, value = mean ± s.d., two-tailed, unpaired $t$-test, *$p = 0.012$). **c** Dashed lines showing the diameter of a control (left) and treated (right) organoid averaged to determine the diameter. **d** Interconnectivity of chambers measured by their separation by thin TNNT2 + filaments or by thin channels showing clear connection ($n = 10$ organoids per condition, value = mean ± s.d., two-tailed, unpaired $t$-test, *$p = 0.041$). **e** immunofluorescence images of hHOs showing interconnected chambers (yellow arrows), TNNT2 + filaments (white arrows), and channels connecting chambers (green arrows), DAPI (blue), TNNT2 (red), scale bar: 500 μm, inset: 100 μm. **f** PECAM1 + cells as a percentage of total DAPI + nuclei, ($n = 7$ organoids per condition, value = mean ± s.d., two-tailed, unpaired $t$-test, **$p = 0.0023$). **g** Immunofluorescence images of hHOs showing DAPI (blue), PECAM1 tissue (green), and TNNT2 tissue (red), scale bar: 500 μm, inset: 50 μm ($n = 12$).

droplets, suggesting dysfunctional lipid metabolism and a more glycolytic profile (Fig. 7f, Supplementary Fig. 8e). None of these phenotypes were found in NHOs. Compared with normal glycemia conditions, diabetic hHOs showed decreased MYL2 + ventricular cardiomyocytes and enlarged MYL7 + atrial cardiomyocyte regions,

indicative of structural defects such as those observed in CHD (Fig. 7g, Supplementary Fig. 8f). Immunofluorescence for myocardial and epicardial markers revealed a drastic difference in the morphological organization as PGDHOs contained epicardial tissue surrounded by myocardial tissue, whereas NHOs contained epicardial

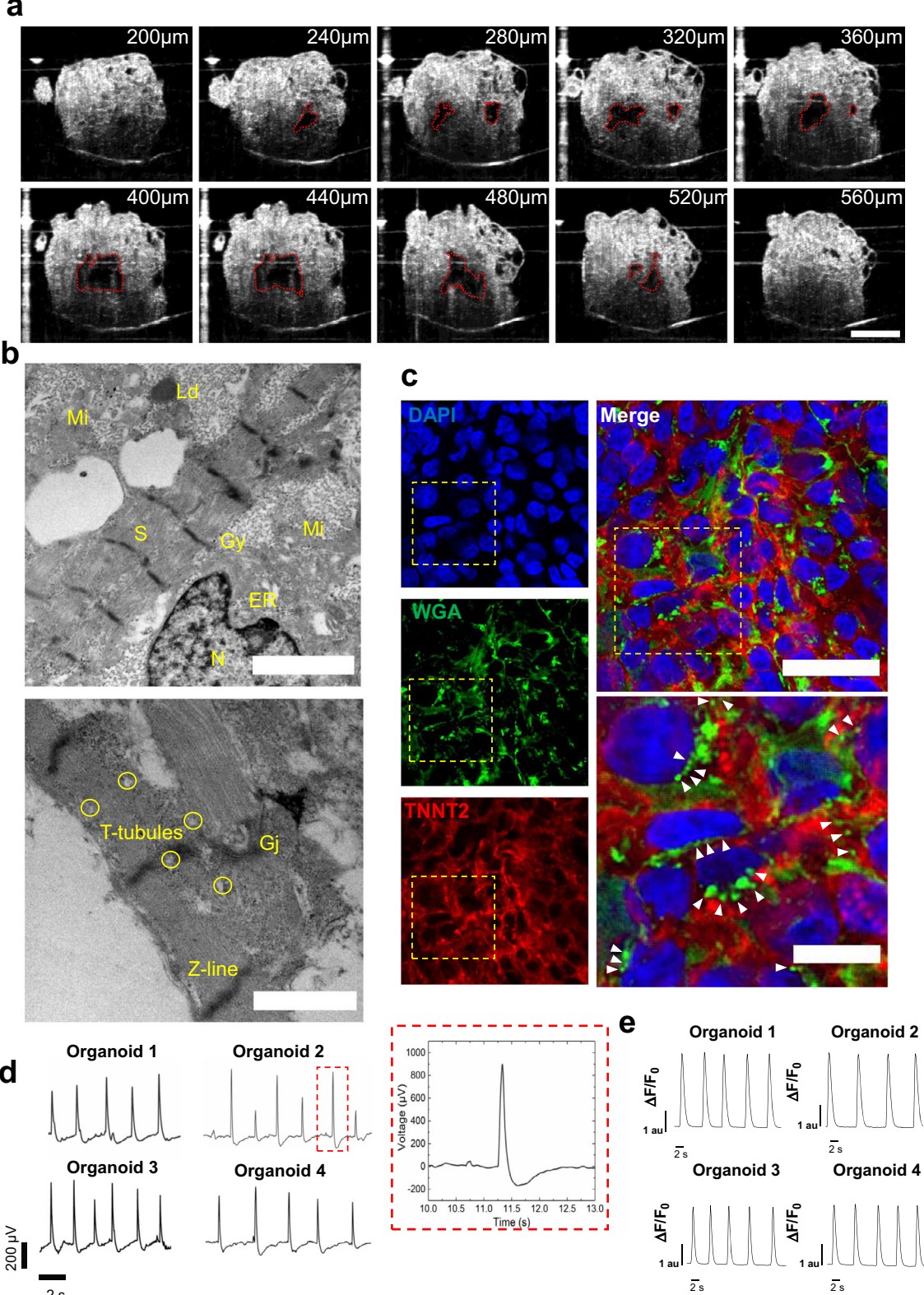

**Fig. 6 Heart organoids recapitulate functional and structural features of the developing heart. a** Optical coherence tomography images showing cross-sections through an organoid, revealing chambers; scale bar: 500 μm. **b** TEM images of hHOs showing endoplasmic reticulum (ER), gap junctions (Gj), glycogen granules (Gy), lipid droplets (Ld), mitochondria (Mi), nucleus (N), and sarcomeres (S); scale bars: 2 μm (top), 1 μm (bottom). **c** Immunofluorescence images of myocardial tissue in hHOs showing WGA staining of T-tubule-like structures (green); white arrowheads indicate representative T-tubule-like structures between cardiomyocytes; scale bar: 50 μm, inset: 20 μm. **d** Electrophysiology recordings of 4 organoids on microelectrode array spanning 15 s and a representative action potential wave (inset). **e** $Ca^{2+}$ transients in 4 representative hHOs after two weeks of differentiation.

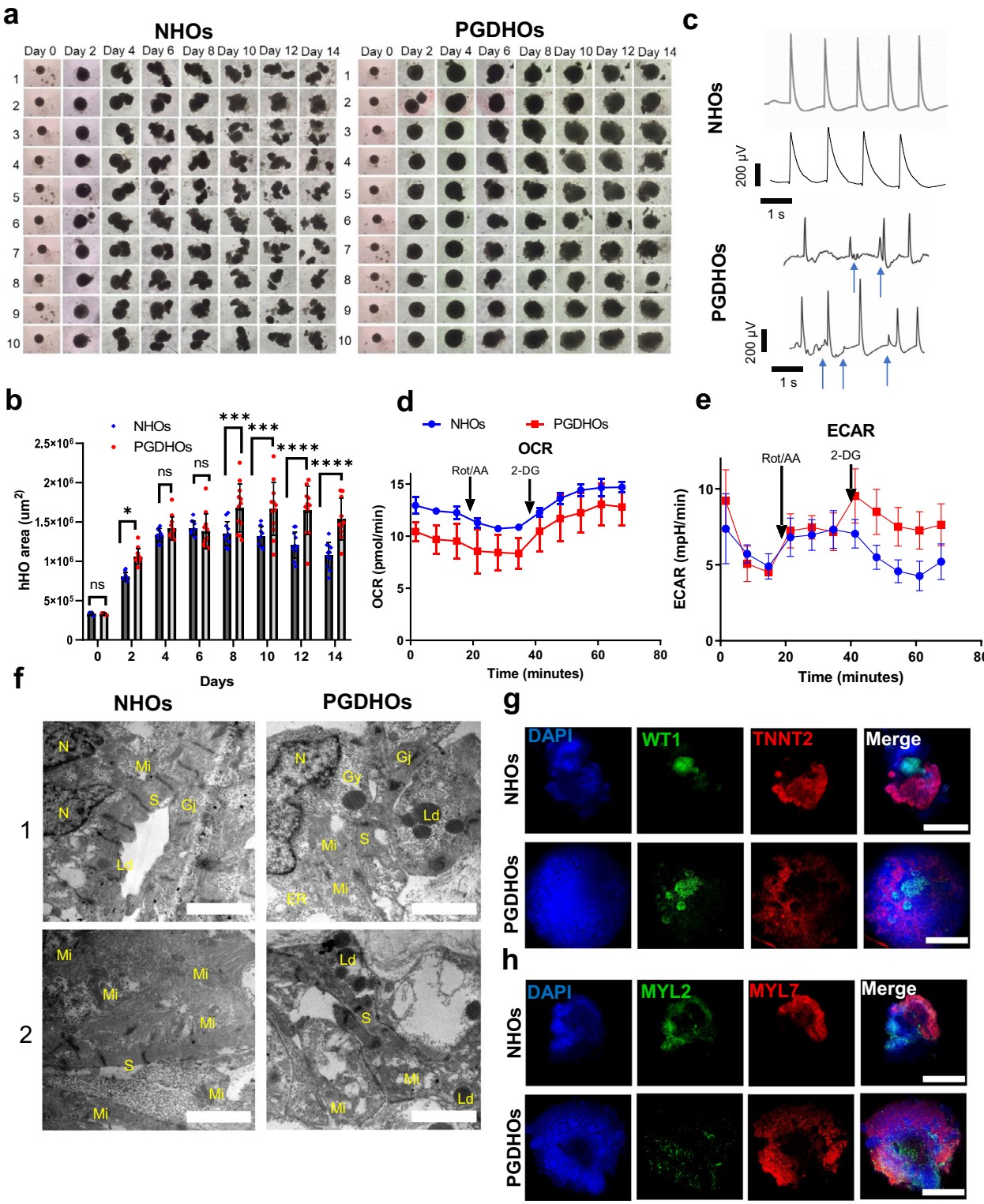

tissue on top of or beside myocardial tissue as physiologically expected (Fig. 7h). These impairments in structural/developmental organization and lipid metabolism in PGDHOs are consistent with expected phenotypes found in PGD-induced CHD. Taken together, our data suggest significant molecular and metabolic perturbations between NHOs and PGDHOs consistent with previous studies on PGD suggesting increased oxidative stress, cardiomyopathy, and altered lipid profiles[48–50], constituting a significant step forward towards modeling metabolic disorders in human organoids.

## Discussion

In recent years, hPSC-derived cardiomyocytes have become critically useful tools to model aspects of heart development[12,26,51], human genetic cardiac disease[52–55], therapeutic screening[14,56], and cardiotoxicity testing[57–60]. Nonetheless, the complex structural morphology and multitude of tissue types present in the human heart impose severe limitations to current in vitro models. Previous attempts at generating 3D human cardiac tissues typically included cardiomyocytes and only one or two other cardiac

**Fig. 7 Human heart organoids faithfully recapitulate hallmarks of pregestational diabetes-induced congenital heart disease. a** Brightfield images following the development of 10 hHOs under normal glycemic conditions (NHOs, left) and under diabetic conditions (PGDHOs, right) over two weeks of differentiation. **b** Area of hHOs under a light microscope over the first two weeks of differentiation (value = mean ± s.d.; $n = 12$; 2-way ANOVA Sidak's multiple comparisons test, exact $p$-values: day 0: $p > 0.99$, day 2: $p = 0.017$, day 4: $p = 0.94$, day 6: $p = 0.99$, day 8: $p = 0.0008$, day 10: $p = 0.0003$, days 12 and 14: $p < 0.0001$). **c** Electrophysiology was performed on NHOs and PGDHOs at 15 days. Arrows indicate arrhythmic events. **d** Seahorse analysis for oxygen consumption rate (OCR), and **e** extracellular acidification rate (ECAR) of normal and diabetic hHOs (Value = mean ± s.d; $n = 3$ organoids per condition). **f** Ultrastructural analysis by TEM of NHOs and PGDHOs showing endoplasmic reticulum (ER), gap junctions (Gj), glycogen granules (Gy), lipid droplets (Ld), mitochondria (Mi), nucleus (N), and sarcomeres (S); scale bars: 2 μm. **g** Confocal immunofluorescence images at differentiation day 15 for cardiac (TNNT2, red) and epicardial (WT1, green) formation, scale bar: 500 μm. **h** Confocal imaging for ventricular (MYL2, green) and atrial (MYL7, red) chamber formation under normal and diabetic-like conditions, scale bar: 500 μm. 2-DG: 2-deoxy-D-glucose, ECAR: extracellular acidification rate, NHOs: normal heart organoids, OCR: oxygen consumption rate, PGDHOs: pregestational diabetes heart organoids, Rot/AA: Rotenone and Antimycin A. Source data are provided as a Source Data file.

cell lineages[61–63]. Here, we sought to create a highly reproducible, scalable, and cost-effective differentiation protocol that yields physiologically relevant human heart organoids with high structural and multicell type complexity using hPSCs. We created and optimized multistep manipulation conditions for canonical WNT signaling using GSK3 and *PORCN* inhibitors in multiple PSC lines. These conditions lead to the formation of most cardiac lineages in a self-assembling heart organoid with similar properties to the fetal heart. This method consistently yields cardiac organoids comprised of approximately 59% cardiomyocytes, 16% epicardial cells, 14% endocardial cells, 12% cardiac fibroblasts, and 1.6% endothelial cells and shows robust beating throughout the entire structure within a week from differentiation initiation and up to at least 8 weeks in culture (longer culture times were not attempted). The organization and specification of these cell types may be related to HAND transcription factor expression, as HAND1 and HAND2 lineage-derived cells contribute to the developing myocardium, epicardium, endocardium, and vasculature[37,64–66]. The fact that both FHF and SHF HAND markers are present suggests that they could play a role in the development of the structural and cell type complexity seen in our hHOs. Notably, hHOs were successfully derived from four independent iPSC lines and one ESC line, demonstrating reproducibility. When compared with existing cardiomyocyte monolayer differentiation methods, hHOs showed higher expression of genes associated with conduction, contractile function, calcium handling, and various cardiac cell populations, which better resemble gene expression data retrieved from human fetal hearts. The depiction of a complex transcriptome highly recapitulative of human fetal heart tissue further strengthens the potential use of hHOs as models of human heart development.

The epicardium, an epithelial layer that encapsulates the human heart, is involved in many important heart processes, including heart development, metabolism, lipid homeostasis, and myocardial injury responses[67,68]. Epicardial signaling cascades are essential for cardiac lineage specification[67]. During embryonic development, cells from the proepicardial organ (PEO), an extracardiac cluster of embryonic cells[68], migrate to the surface of the heart to form the epicardium. Some of these cells can undergo epithelial-mesenchymal transition (EMT) to generate other cardiac lineages including cardiac fibroblasts[36,67–69]. Due to its capacity to communicate with the myocardium and its ability to mobilize stem cell populations, the epicardium has become a key focus of research in cardiac regeneration and repair[22,67,68]. The epicardium also plays a fundamental but underexplored role in multiple types of cardiovascular and metabolic disease, including diabetic cardiomyopathy, coronary artery disease, and metabolic syndrome. In this last condition, epicardial-derived fat experiences a significant expansion and correlates strongly with morbidity, highlighting the potential relevance of the epicardium to

human disease. To increase the complexity of our system, and inspired by a previous epicardial monolayer differentiation method[22], we created and optimized conditions for producing heart organoids with well-defined regions of epicardial tissue adjacent to myocardial tissue. These epicardial-myocardial interactions are important in mammalian heart development and function, as epicardial cells increase cardiomyocyte growth in 3D engineered heart tissues, and co-transplantation of both cell types into rat hearts increases endothelial cell production[63]. Our hHO protocol will facilitate the study and modeling of epicardial-myocardial interactions in vitro.

Together with the use of small-molecule inhibitors that manipulate canonical Wnt signaling pathways, successful cardiomyocyte differentiation has been achieved in the past using morphogens such as BMP4 and Activin A[21,26]. These growth factors lead to the induction of cardiac mesoderm in the embryo[70] and established differentiation protocols using them show effective differentiation to various cardiac mesoderm progenitors[23,70,71]. Recently, gradient exposures to specific concentrations of BMP4 and Activin A have been studied in the specification of first and second heart field formation[16]. The addition of these growth factors to the initial CHIR exposure in our hHO differentiation protocol led to improved morphological features, such as increased chamber interconnectivity and vascularization.

The important role that cardiac fibroblasts play on cardiac development and cardiac matrix production/organization is often overlooked in in vitro models. Most cardiac fibroblasts in embryonic development arise from the PEO[36,72,73], highlighting the necessity of epicardial induction in developmental heart models. These fibroblasts facilitate cardiomyocyte functionality in hPSC-derived 3D cardiac microtissues, and as such, their inclusion in any in vitro human heart model is paramount[61]. Immunofluorescence analysis of our hHOs revealed the presence of cardiac fibroblast markers including the membrane glycoprotein THY1, which is involved in cell-cell and cell-matrix adhesion[73,74], and the intermediate filament protein Vimentin, typically seen in cells of mesenchymal lineage[74]. Other cardiac fibroblast markers were found in the hHOs via RNA-sequencing analysis, including *DDR2* which plays an important role in EMT[73], and the FHF marker *PDGFRα*, which is also crucial for vascularization during development[74]. These data provide a strong indication of the increased complexity of our hHO system and its close resemblance to fetal heart tissue. It should be noted that while our immunofluorescence and bulk RNA sequencing data suggest the presence of the cardiac cell types described above, these techniques may not fully capture the heterogeneity of our organoids. In recent years, significant advances have been made in single-cell and single-organoid "omics" technologies[75–77] as well as the ability to image 3D engineered tissues with unprecedented quality

and resolution[9,78–80]. Further studies employing these techniques will add significantly more detail and information to the cell type compositions at different time points of differentiation.

An acute limitation of many organoid systems is a lack of a functional vascular network to facilitate the exchange of nutrients and removal of waste material, as they instead rely solely on diffusion[4,81,82]. Several vascularized organoids have been described in the literature modeling the brain[4], kidney[83], and blood vessel[84], however, none have been described modeling the heart. In these studies, various techniques are used to induce vascularization including implantations in mice[4], culturing the organoids under flow[83], and embedding endothelial cells in a Matrigel/collagen matrix, and inducing their migration[84] to create a vascular network. Remarkably, we observed the formation of vascular structures in our final protocol for hHOs without any additional steps. We also observed a point of high vascular branching that merits future exploration into the developmental point of origin in heart vascularization. Further studies into the functionality of this vascular tissue will be necessary, particularly to determine the maturity of the vessels, their levels of connectivity, and if they closely resemble coronary vasculature. This latter feature would open the door to modeling coronary vasculature pathologies that arise due to CVD and metabolic disorders.

In addition to endothelial structures, we also observed spontaneous hHO reorganization into interconnected chambers, a powerful 3D feature that recapitulates fetal-like organogenesis. Previous studies of microchamber formation in vitro utilized micropatterning of hPSCs into a confined area to generate 3D cardiac microchambers with cell-free regions, a myofibroblast perimeter, and nascent trabeculae[15]. Other reports have produced 3D bio-printed hPSC-laden scaffolds and differentiated them to beating cardiac microtissues with two chambers[85]. While the structures generated in these studies showed some fetal-like formation of cardiac microchambers, they lacked endocardial tissue[25], a crucial player in heart maturation and morphogenesis[86]. The hHOs reported here form multiple chambers lined with NFATC1+ endocardial cells which are interconnected as seen in the OCT cross-sectional imaging (Supplementary Movies 7–10). Expression of specific ECM genes in the hHOs resembling the fetal heart matrix, such as *COL1A1*, *COL4A1*, *COL5A2*, *FBN1*, *EMILIN1*, *HSPG2*, and *LAMA2* (Supplementary Fig. 3d) might be an important factor in chamber organization, as ECM components have been shown to mediate the formation of chambered mouse cardiac organoids[18]. Therefore, the expression of these genes in our hHOs deserves further examination in the future. The chambers may also specify further into atrial-like and ventricular-like regions, as cardiomyocytes from both lineages are seen in separate regions in our hHOs.

In the past few years, 3D human cardiac tissues have been used to model genetic and non-genetic conditions (myocardial infarction, drug cardiotoxicity)[9,87]. We provide evidence that heart organoids can be valuable models to study CHD in pregestational diabetes-like conditions. Maternal diabetes is one of the most common causes of newborn CHD (up to 12% of newborns from diabetic mothers have some form of CHD[88]). Using healthy and diabetic levels of glucose and insulin in the differentiation media, we demonstrate the effects of diabetic conditions on the developmental process of hHOs. Organoids developing in healthy conditions displayed active structural changes including patterning, while hHOs in diabetic conditions developed largersizes reminiscent of macrosomia. The larger size of diabetic hHOs also suggests potential signs of cardiac hypertrophy, a hallmark of maternal PGD[89]. We also observed a reduction in mitochondria, dysfunctional lipid metabolism, and impaired structural organization. These data suggest hHOs might be useful tools for the study of PGD-induced CHD, and might pave the way for the identification of pharmacological agents aimed at treating or preventing this condition.

Although our technology offers exciting opportunities to model human congenital heart disease in vitro, significant limitations still exist. First, organoids tend to deviate from their normal developmental pathway as a function of time, becoming less relevant the longer they are cultured. Second, their ability to recapitulate heart development is still limited when compared to other existing models, such as mice, even though they have the significant advantage of being human in origin rather than a surrogate animal model. There is large room for improvement in the technology, particularly in trying to better recapitulate morphological and anatomical features and inducing the formation of effective vascular networks that can provide nutrients.

In summary, we describe here a highly reproducible and high-throughput human heart organoid generation method relying on self-assembly triggered by developmental cues and provide proof-of-concept for the modeling of a congenital heart disorder. Heart organoids present multicell type and morphological complexity reminiscent of the developing human fetal heart, including chamber formation, atrioventricular specification, electro-physiological activity and vascularization. Heart organoids can be used to model features of pregestational diabetes-induced congenital heart disease and might thus constitute useful models for the study of the molecular pathology of congenital heart disease in humans in the future.

## Methods

**Stem cell culture**. The following human iPSC lines were used in this study: iPSC-L1, AICS-0037-172 (Coriell Institute for Medical Research, alias AICS), iPSCORE_16_3 (WiCell, alias iPSC-16; UCSD013i-16-3), iPSC GCaMP6f[43,44], and human ESC line H9 (WiCell, WA09). All PSC lines were validated for pluripotency and genomic stability. hPSCs were cultured in Essential 8 Flex medium containing 1% penicillin/streptomycin (Gibco) on 6-well plates coated with growth factor-reduced Matrigel (Corning) in an incubator at 37 °C, 5% $CO_2$ until 60–80% confluency was reached, at which point cells were split into new wells using ReLeSR passaging reagent (Stem Cell Technologies).

**hPSC monolayer cardiac differentiation**. Differentiation was performed using the small molecule Wnt modulation strategy adapted from a previous protocol[21] (referred to as monolayer 1 in the text), with small modifications. Briefly, differentiating cells were maintained in RPMI with B27 minus insulin from day 0–7 of differentiation and RPMI with B27 supplement (Thermo) from day 7–15 of differentiation. Cells were treated with 10 μM CHIR99021 (Selleck) for 24 h on day 0 of differentiation and with 2 μM Wnt-C59 (Selleck) for 48 h from day 3–5 of differentiation.

**Self-assembling human heart organoid differentiation**. A step-by-step protocol describing the fabrication and differentiation of the human heart organoids can be found at Protocol Exchange[90]. Accutase (Innovative Cell Technologies) was used to dissociate PSCs for spheroid formation. After dissociation, cells were centrifuged at 300 *g* for 5 min and resuspended in Essential 8 Flex medium containing 2 μM ROCK inhibitor Thiazovivin (Millipore Sigma). hPSCs were then counted using a Moxi Cell Counter (Orflo Technologies) and seeded at 10,000 cells/well in round bottom ultra-low attachment 96-well plates (Costar) on day −2 at a volume of 100 μl per well. The plate was then centrifuged at 100 *g* for 3 min and placed in an incubator at 37 °C, 5% $CO_2$. After 24 h (day −1), 50 μl of media was carefully removed from each well, and 200 μl of fresh Essential 8 Flex medium was added for a final volume of 250 μl/well. The plate was returned to the incubator for an additional 24 h. On day 0, 166 μl (~2/3 of total well volume) of media was removed from each well and 166 μl of RPMI 1640/B-27, minus insulin (Gibco) containing CHIR99021 (Selleck) was added at a final concentration of 4 μM/well along with BMP4 at 0.36 pM (1.25 ng/ml) and Activin A at 0.08 pM (1 ng/ml) for 24 h. On day 1, 166 μl of media was removed and replaced with fresh RPMI1640/B-27, minus insulin. On day 2, RPMI/B-27, minus insulin, containing Wnt-C59 (Selleck) was added for a final concentration of 2 μM Wnt-C59 and the samples were incubated for 48 h. The media was changed again on day 4 and day 6, but insulin was not added to the RPMI1640/B-27 (Gibco) mixture until day 6, because it has been shown to decrease cardiomyocyte yield before this time point[21]. On day 7, a second 2 μM CHIR99021 exposure was conducted for 1 h in RPMI1640/B-27. Subsequently, media was changed every 48 h until organoids were ready for analysis. Diabetic conditions were simulated by using basal RPMI media with 11.1 mM glucose and 1.14 nM insulin and compared with control media containing 3.5 mM glucose and 170 pM insulin. 40.5 μM Oleate-BSA (Sigma), 22.5 μM Linoleate-BSA(Sigma), and 120 μM L-Carnitine (Sigma) were added at day 7 to increase fatty acid metabolism, concentrations previously described[34].

**Lentiviral transduction**. For lentiviral production HEK293T (Horizon Inspired Cell Solutions, HCLXXXX) cells were transfected with the Flip-GFP plasmid (VectorBuilder) and the packaging plasmids (pMD2; psPAX2) using lipofectamine with Plus reagent (Thermo). Lentivirus was added to iPSC-L1 cells with 8 µg/ml polybrene (Fisher Scientific) and incubated overnight. Puromycin selection was carried out for ~3–5 days. Surviving clones were collected, replated, and expanded to give rise to the FlipGFP line.

**Immunofluorescence**. hHOs were transferred to microcentrifuge tubes (Eppendorf) using a cut 1000 µL pipette tip to avoid disruption to the organoids and fixed in 4% paraformaldehyde solution. Fixation was followed by washes in PBS-Glycine (20 mM) and incubation in blocking/permeabilization solution containing 10% Donkey Normal Serum, 0.5% Triton X-100, 0.5% bovine serum albumin (BSA) in PBS on a thermal mixer (Thermo Scientific) at minimum speed at 4 °C overnight. hHOs were then washed 3 times in PBS and incubated with primary antibodies (Suppl. Table 1) in Antibody Solution (1% Donkey Normal Serum, 0.5% Triton X-100, 0.5% BSA in PBS) on a thermal mixer at minimum speed at 4 °C for 24 h. Primary antibody exposure was followed by 3 washes in PBS and incubation with secondary antibodies (Supplementary Table 1) in Antibody Solution on a thermal mixer at minimum speed at 4 °C for 24 h in the dark. T-tubules staining was conducted using Wheat Germ Agglutinin (WGA) lectins conjugated with FITC (Millipore Sigma). The stained hHOs were washed 3 times in PBS before being mounted on glass microscope slides (Fisher Scientific) using Vectashield Vibrance Antifade Mounting Medium (Vector Laboratories). 90 µm Polybead Microspheres (Polyscience, Inc.) were placed between the slide and the coverslip (No. 1.5) to preserve the 3D structure of the organoids.

**Confocal microscopy and image analysis**. Samples were imaged using confocal laser scanning microscopy (Nikon Instruments A1 Confocal Laser Microscope; Zeiss LSM 880 NLO Confocal Microscope System). Images were analyzed using Fiji (https://imagej.net/Fiji). For cell quantification in the organoids, DAPI positive cells were counted and used for normalization against the target cell marker of interest across at least three z-planes throughout each organoid for each target cell marker.

**RNA sequencing and transcriptomic analysis**. RNA was extracted at 11 different time points throughout the hHO fabrication and differentiation protocol shown in Fig. 1a. The time points are as follows: days 0, 1, 3, 5, 7, 9, 11, 13, 15, 17, and 19. At each time point, eight organoids were removed and stored in RNAlater (Qiagen) at −20 °C until all samples were collected. RNA was extracted using the Qiagen RNEasy Mini Kit according to manufacturer instructions (Qiagen), and the amount of RNA was measured using a Qubit Fluorometer (Thermo). RNA samples were sent to the MSU Genomics Core, where the quality of the samples was tested using an Agilent 2100 Bioanalyzer followed by RNA sequencing using an Illumina HiSeq 4000 system. For RNA-seq sample processing, a pipeline was created in Galaxy. Briefly, sample run quality was assessed with FASTQC, and alignment to hg38 was carried out using HISAT2. Counts were obtained using featureCounts and differential expression analysis was performed with EdgeR. Further downstream bioinformatic analysis was performed in Phantasus 1.11.0 (artyomovlab.wustl.edu/phantasus) and ToppGene Suite (http://toppgene.cchmc.org).

**Optical coherence tomography analysis**. A customized spectral-domain OCT (SD-OCT) system was developed to acquire 3D images of the cardiac organoids. As shown in Suppl. Fig. 9, a superluminescent diode (SLD 1325, Thorlabs) was used as the light source to provide broadband illumination with a central wavelength of 1320 nm and a spectral range of 110 nm. The output of the SLD was split 50/50 with a fiber coupler and transmitted to the sample and reference arms, respectively. A galvanometer (GVSM002-EC/M, Thorlabs) was used to scan the optical beam in transverse directions on the sample. The SD-OCT setup used a custom-designed spectrometer consisting of a 1024-pixel line scan camera (SU1024-LDH2, Sensors Unlimited), an 1145-line pairs per mm diffraction grating (HD 1145-line pairs per mm at 1310 nm, Wasatch Photonics), and an $f = 100$ mm F-theta lens (FTH100-1064, Thorlabs). The sensitivity of the OCT system was measured as ~104 dB when operating at a 20 kHz A-scan rate. The axial resolution of the SD-OCT system was measured to be ~7 mm in tissue. A 5X objective lens (5X Plan Apo NIR, Mitutoyo) was used to achieve a transverse image resolution of ~7 mm, and the scanning range used for the cardiac organoids imaging was ~2 mm × 2 mm. Fixed hHOs were placed into a 96-well plate with PBS and imaged at a 20-kHz A-scan rate. Obtained OCT datasets of the cardiac organoids were first processed to generate OCT images with corrected scales. Then OCT images were further de-noised using a speckle-modulation generative adversarial network[91] to reduce the speckle noise. 3D renderings of OCT images were performed using Amira software (Thermo Fisher Scientific).

**TEM sample preparation and imaging**. Organoids were fixed in 4% PFA for 30 min followed by 3 washes in water, 10 min each. Post fixation was performed in 1% osmium tetroxide in cacodylate buffer (pH 7.3) for 60 min at room temperature. Organoids were embedded in 2% agarose in water, solidified using ice, for manipulation. Then, a serial dilution of acetone was used for dehydration (25%, 50%, 75%, 90%, and 3 times in 100%) for 10 min each. Organoids were infiltrated with Spurr resin (Electron Microscopy Sciences) by immersion in 1:3, 2:2, and 3:1 solutions of resin in acetone, 3 h each under agitation, following embedding in 100% resin for 24 h, and polymerization at 60 °C overnight. Ultra-thin sections (50–70 nm) were cut using RMC PTXL Leica Ultramicrotome and collected in carbon-coated copper grids 200 mesh. Before observation, all samples were positively stained in 2% uranyl acetate and 1% lead citrate for 6 and 3 min, respectively. The grids were examined at 100 keV using a JEOL 1400 Flash transmission electron microscope.

**Electrophysiology**. An in-house microelectrode array (MEA) system described previously[92] was used to record the electrical activity of individual organoids (Supplementary Fig. 7). The Microelectrode array (MEA) was fabricated with the following cleanroom procedures. First, 10 µm Parylene C was deposited on a cleaned 3-inch silicon wafer (PDS 2010, Specialty Coating System, Inc). Then, 500 nm Au was evaporated on the substrate. Next, a photoresist (PR) layer was spun on Au and photolithographically patterned the areas of 32-channel microelectrodes, interconnection wires, and contact pads. Finally, 2 µm Parylene C was deposited on the substrate as an insulating layer, and then Parylene C on the contact pads and microelectrodes were removed completely using oxygen plasma dry etching (RIE-1701 plasma system, Nordson March, Inc). Live organoids were placed on the MEA inside a PDMS well in culture media supplemented with 15 mM HEPES. The MEA was placed within a Faraday cage inside an incubator at 37 °C at low humidity to avoid damage to the MEA system. Each organoid was recorded for a period of 30 min, and the PDMS well was washed with PBS between organoids. The recorded signals were amplified and digitalized using a commercial Intan RHD2132 system (Intan Technologies) and then recorded with Intan RHD2000 interface and analyzed using the Matlab Chronux toolbox to extract the electrocardiogram (ECG) from the recordings.

**Calcium transient analysis**. Calcium transients were observed in heart organoids developed from iPSCs expressing the fast calcium indicator GCaMP6f[43,44]. Dynamic fluorescence changes in the heart organoid were recorded at 10 frames per second on an inverted fluorescence microscope (IX71, Olympus). Data analysis of fluorescence recordings was performed in MATLAB. 10 by 10 pixel binning was applied to the fluorescence recordings to minimize impact of contraction of heart organoids. Baseline $F_0$ of the fluorescence intensities $F$ was calculated using asymmetric least squares smoothing[93]. Fluorescence change $\Delta F/F_0$ was calculated by:

$$\frac{\Delta F}{F_0} = (F - F_0)/F_0 \qquad (1)$$

**Seahorse metabolic analysis**. A Seahorse analyzer (Agilent) was used to conduct glycolysis rate assay as per manufacturer instructions. Organoids were carefully dissociated using STEMDiff cardiomyocyte dissociation kit and checked for viability using a hemocytometer before analysis. Only samples with over 90% viability were used in the assay. Identical numbers of cells were plated for the assay. Measurements were performed immediately after dissociation.

**Statistics and reproducibility**. All analyses were performed using GraphPad software and all raw data was collected in Microsoft Excel. All data presented a normal distribution. Statistical significance was evaluated with a standard unpaired Student $t$-test (2-tailed; $P < 0.05$) when appropriate. For multiple-comparison analysis, 1-way ANOVA with the Tukey's or Dunnett's post-test correction was applied when appropriate ($P < 0.05$). All data are presented as mean ± s.d. and represent a minimum of 3 independent experiments with at least 3 technical replicates unless otherwise stated. All micrograph images are representative of at least 6 independent experiments per condition/marker and calcium transient graphs are representative of 6 independent experiments.

**Reporting summary**. Further information on research design is available in the Nature Research Reporting Summary linked to this article.

## Data availability

The organoid RNA-Sequencing data sets have been deposited in the National Center for Biotechnology Information Gene Expression Omnibus repository under accession code "GSE153185". The RNA-seq from monolayer differentiation method 2 and fetal heart data were obtained from the National Center for Biotechnology Information Gene Expression Omnibus repository under accession code "GSE106690"[43]. The raw data for all graphs generated in this study are provided in the Supplementary Information/Source Data file. All other data generated and/or analyzed in this study are provided in the published article and its supplementary information files or from the corresponding author upon reasonable request. Source data are provided with this paper.

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

## Acknowledgements

We wish to thank the MSU Advanced Microscopy Core and Dr. William Jackson at the MSU Department of Pharmacology and Toxicology for access to confocal microscopes, and the MSU Genomics Core for sequencing services. We also wish to thank all members of the Aguirre Lab for their valuable comments and advice. Work in Dr. Aguirre's laboratory was supported by the National Heart, Lung, and Blood Institute of the National Institutes of Health under award numbers K01HL135464, R01HL151505, by the American Heart Association under award number 19IPLOI34660342 and by the Spectrum-MSU Foundation. Work in Dr. Zhou's laboratory was supported by grants from the National Institutes of Health under award number R01EB025209. Work in Dr. Li's laboratory was supported in part by the National Science Foundation under award number ECCS-2024270 and Michigan State University Graduate Excellence Fellowship.

## Author contributions

Y.L.I. and A.A. designed all experiments and conceptualized the work. Y.L.I. performed all experiments and data analysis. A.H.W. performed cell and organoid culture and confocal imaging. M.A.G. performed cell and organoid culture. K.A.B. performed cell culture. B.D.V. created cell lines for apoptosis analysis. W.Y. and W.L. designed the electrophysiological instruments, conducted electrophysiology recordings, and data analysis. J.Z., G.N., Y.M. and C.Z. performed optical coherence tomography experiments and data analysis. Y.M. performed calcium imaging. N.P. and X.C. performed TEM sample preparation and imaging. Y.L.I. created Figs. 1a, i, 3g, Suppl. Fig. 8a, Y.M. created Suppl. Fig. 9. Y.L.I., A.H.W. and A.A. wrote the manuscript.

## Competing interests

The authors declare no conflicts of interest.
