## [Peer Review File · Nature Communications]

Reviewers' Comments:

Reviewer #1:

Remarks to the Author:

This is a revised manuscript that was initially submitted to NBE, where authors reported an improved protocol to generate heart organoids using hPSC-derived cells. The main concerns from this reviewer are 1) manuscript is too technical, lacks of new biological insights; 2) lack of data to determine the heterogeneity of their organoids (this reviewer recommended single cell omics approach); 3) lack of experimental validation after genomics data analysis; 4) need of additional functional data to support successful generation of various cardiac cell types within the organoids. Now authors have made the following efforts:

Explanation was provided to clarify the novelty of this manuscript to some extent (no new data here though) and how the protocol is different from previously published ones.

Authors made valid points on why using bulk RNA-seq instead of scRNA-seq. After careful evaluation of their current presentation of bulk RNA-seq data, this reviewer believes it is acceptable, especially in conjunction with all other data.

Authors have now included additional data from various assays to better characterize the cell populations within the organoids. Immunofluorescence images, additional OCT images and new quantifications all help improve the manuscript.

This reviewer appreciates the significance of their study, the high quality of their data and the responsiveness to previous comments. Now this reviewer has only two remaining comments for authors to address prior to the acceptance of this manuscript for publication:

1. It will be helpful to acknowledge the inherent caveats of using such organoids and tone down the conclusions. This is critical given the general concern from the field, which was indeed reflected from reviewers' comments during the initial round of review at NBE.
2. It is recommended to discuss the increasing heterogeneity within organoids and future efforts using integrative single cell multi-omics coupled with high-resolution imaging and in situ bar-coding to address such heterogeneity. With the quickly evolved genomics and imaging technology, the sequencing depth and spatial resolution have been much improved. Authors can refer to some latest publications to discuss about some future directions.

A minor point: please avoid using "him" to refer a particular reviewer in the rebuttal letter. The time and efforts spent could from a reviewer with different sex.

Evaluations on responses to Reviewer #3:

Most technical questions were addressed adequately by either correcting the wrong information or clarifying with additional explanation.

This reviewer noticed that when authors responded to Reviewer #3's concern "Is there any significance to the fact that HAND1 cytoplasmic staining appears to be restricted to a particular region of the organoid, whilst mostly nuclear in the other regions?", they only explained rather than providing new improved images or at least several representative images to support their explanation rather than leaving the wrong impression to readers as did to Reviewer#3.

Other minor comments and concerns have been addressed well too.

Reviewer #2:

Remarks to the Author:

Reviewer #2

According to this reviewer the comments have not been adequately addressed by the authors. They failed to demonstrate the key advantages and use of their new model. Also many results are preliminary and need further expanding, depth and quality. The model of diabetes seems promising as it is one of not many models in this field and addresses the fetal cardiac response specifically, but it is preliminary especially and some experimental set-ups are even flawed.

As mentioned in the Reviewer #2's Summary the presented model of diabetes needs to be clearly distinguishable and of better use from others, such as 2D models. The authors haven't compared their model to any already present ones nor described what makes their model better to use.

The authors mention no benefits of using BMP+Activin A protocol as compared to CHIR induction, yet this is not presented or compared. Only a comparison of CHIR with CHIR+BMP+Activin A is shown in Fig 2, but BMP+Activin A alone may yield similar results without CHIR. This data should be presented.

Also, the authors need to present functional inhibition of endogenous morphogens or remove these speculations from the text.

The cell type analyses all require quantification and not using the % area. Flow cytometry, single nuclei RNAS-seq and immunostaining of nuclei (eg. NKX2-5) are all options.

I agree that TEMs are gold standard for t-tubules, but the presented images are not convincing. Neither are the new presented analyses using WGA staining.

The increased number of HAND1/HAND2 expression images prove spatiotemporal progress and repeated occurrence, nevertheless the authors haven't address the ventricular/atrial spatial compartmentalization reproducibility. Would the sectioning of those two areas always be this distinctive within the hHO and in the same area?

Regarding electrophysiology in Figure 6d, the explanation given by the authors clearly states the limitations of the system used and prevents the authors from obtaining good quality quantifiable data. Therefore, it is impossible to determine whether there is a difference between the organoids under diabetic conditions or whether they don't attach as well to the MEA system (or even if they are attached to a different region as the organoids are heterogeneous). All this data is therefore flawed and cannot be used.

To prove a point of presence of interconnecting chambers and vessels and through that a lack of hypoxic regions, proper O₂ diffusion should be determined. For example, staining with pimonidazole could be used to distinguish hypoxic regions within hHO. The possibility of O₂/hypoxia in larger PGDHOs or other possible explanations have not been adequately addressed.

I thank the authors for including the new methods for the number of cells for Seahorse assays. However, I appear to have overlooked that the cells were dissociated, rather than organoids used for the assays. This is therefore flawed as the cells will behave very differently in 2D vs 3D and there may also be substantial skewing of the data due to the dissociation/survival/attachment of the cells. Therefore, I'm not sure this current data accurately reflects the changes in metabolism caused by "diabetes" in the organoids.

The numbering of figures and supplementary figures doesn't match the text in couple places and the numbering of Supplementary Figures is not in order either, making it a bit confusing to refer to.

REBUTTAL LETTER

We want to thank the reviewers for their time and constructive criticism. A detailed point by point rebuttal follows.

Note: Original reviewers' comments have been highlighted in blue for clarity.

Reviewer #1 comments

Major concerns:

It will be helpful to acknowledge the inherent caveats of using such organoids and tone down the conclusions. This is critical given the general concern from the field, which was indeed reflected from reviewers' comments during the initial round of review at NBE.

We have toned down the conclusions as per the reviewer's suggestion and have added a few sentences to clarify downsides of organoid technology and in vitro developmental modeling.

It is recommended to discuss the increasing heterogeneity within organoids and future efforts using integrative single cell multi-omics coupled with high-resolution imaging and in situ bar-coding to address such heterogeneity. With the quickly evolved genomics and imaging technology, the sequencing depth and spatial restoration have been much improved. Authors can refer to some latest publications to discuss about some future directions.

We thank the reviewer for this suggestion. We have now highlighted the potential of single cell approaches and their synergy with organoid technologies. We have also added some references to recent work in this respect.

A minor point: please avoid using "him" to refer a particular reviewer in the rebuttal letter. The time and efforts spent could from a reviewer with different sex.

We apologize for this oversight; we certainly have no assumptions on reviewer's gender or sex.

Reviewer #1 comments regarding Reviewer #3 comments

This reviewer noticed that when authors responded to Reviewer #3's concern "Is there any significance to the fact that HAND1 cytoplasmic staining appears to be restricted to a particular region of the organoid, whilst mostly nuclear in the other regions?", they only explained rather than providing new improved images or at least several representative images to support their explanation rather than leaving the wrong impression to readers as did to Reviewer#3.

We appreciate this comment and have replaced the image in question to a more representative image. We have also added additional representative images to further clarify HAND1 localization (Figure 4 and Suppl. Fig. 5).

Reviewer #2 comments

Summary

They failed to demonstrate the key advantages and use of their new model. Also many results are preliminary and need further expanding, depth and quality. The model of diabetes seems promising as it is one of not many models in this field and addresses the fetal cardiac response specifically, but it is preliminary especially and some experimental set-ups are even flawed.

We disagree with the reviewer's opinion. Our model includes a significant deal of innovation and valuable scientific data, including significant technological advances toward human heart organoid generation with greatly increased complexity when compared to previous protocols. Below is a summary of our main advances:

1) Our new method is completely guided by self-assembly and thus constitutes a close recapitulation of human heart development in vitro. It produces the **most physiologically faithful human heart organoids reported to date** (together with the recently published Hofbauer et al, *Cell* 2021, although our method shows extensive functional data and disease modeling which are absent in Hofbauer et al.). Other human cardiac organoid models reported recently, such as Richards et al., *Nature Biomedical Engineering*, 2020, relied on the manual assembly of cell types from different origins in vitro (e.g. HUVECs, dermal fibroblasts, cardiomyocytes), with significant limitations associated, such as non-cardiac origin cells, lack of minor cardiac cell lineages, and not relevant morphology/structure. Other methods reported in the last few months have used mouse cells (Lee et al., *Nature Communications*, 2020, with the traditional limitations associated to mouse models, or have created gastruloids that contain more than one germ layer and give rise to other type of structures aside from the heart (Drakhlis et al, *Nature Biotechnology* 2021; Rossi et al, *Cell Stem Cell*, 2021). The method we describe **overcomes many of these limitations and present more extensive data, including functional characterization and disease modeling** in humans by using human pluripotent stem cells. Our heart organoids are **unique in their complexity** and naturally contain all major and minor heart lineages (cardiomyocytes, endothelial cells, epicardial cells, cardiac fibroblasts, endocardial cells) in their native organization (including beating chambers with endocardial lining, for example). All of these characteristics allow us to recapitulate early human cardiac development to a degree of detail not seen before and make our organoid system a powerful platform for the study of human development and congenital heart disorders. **To the extent of our knowledge, this is the first report with this wealth of data in such a powerful developmental modeling system.**

2) Our method is a **significant technical step forward** due to the simplicity of its implementation. Any laboratory with standard cell culture equipment can use it, there is no special equipment necessary. Furthermore, our protocol is very **highly reproducible, scalable**, and compatible with **high-throughput** approaches (the current results included in the manuscript represent work on thousands of reproducible individual organoids). These characteristics make it ideal for pharmacological screening, an area where better cardiac models are sorely needed.

3) We provide **proof-of-concept** of the utility of our system for **congenital heart disease modeling** by recapitulating the pathology of pregestational diabetes in congenital heart disease. PGD-induced CHD is a significant medical problem steadily increasing in the developed world. Children of diabetic mothers have up to 12% risk of having CHD, compared to 1% in the normal population. In our heart organoids exposure to PGD conditions leads to phenotypes very similar to those observed in vivo, and thus our platform could be used to identify molecular targets to curtail the progression of this condition. **This is the first time such a complex non-genetic CHD disorder has been modeled in heart organoids successfully.**

4) Finally, we **created novel tools** for the study of heart organoid biology adapting previously existing instrumentation. To characterize and validate our organoid models we created novel in-house instruments for optical coherence tomography (OCT) and electrophysiology recording using flexible μ ECoG arrays.

Major concerns:

As mentioned in the Reviewer #2's Summary the presented model of diabetes needs to be clearly distinguishable and of better use from others, such as 2D models. The authors haven't compared their model to any already present ones nor described what makes their model better to use.

There are no similar models to the one we presented for PDG-induced CHD in humans, this is precisely the value of our model. Our evaluation is based on comparing our data and findings to extensive existing mouse literature and human clinical studies in PGD-induced CHD. In our opinion, a 2D would constitute a step backwards, as it will lose a great deal of complexity and will not be able to model critical aspects of congenital heart defects which typically affect the morphology and structure of the heart. Animal models are out of the scope of this paper.

The authors mention no benefits of using BMP+Activin A protocol as compared to CHIR induction, yet this is not presented or compared. Only a comparison of CHIR with CHIR+BMP+Activin A is shown in Fig 2, but BMP+Activin A alone may yield similar results without CHIR. This data should be presented.

We thank the reviewer for this comment. Since using BMP4 and Activin A alone has been demonstrated by (Andersen et al., *Nature Communications*, 2018), repeating such experiments will be redundant. We observed that our protocol with CHIR alone yields more complex cardiac organoids (containing more cardiac cell types, chambers, and endothelial vessel-like structures) than reported by Andersen et al., 2018, and incorporated the valuable findings of that paper to improve our protocol. Figure 2 demonstrates that the addition of BMP4 and Activin A to our optimized CHIR only protocol improved vascularization, increased organoid size and the interconnectivity of the internal chambers. We will amend the text to better clarify the benefits observed in using CHIR+BMP4+Activin A.

Also, the authors need to present functional inhibition of endogenous morphogens or remove these speculations from the text.

There is significant body of literature on the production of endogenous morphogens in self-assembling organoids, including the recent Hofbauer et al, *Cell* 2021; Drakhlis et al, *Nature Biotechnology*, 2021 and Rossi et al, *Cell Stem Cell*, 2021. Indeed, the production of endogenous morphogens is what drives the self-organization process that mimics development in this context (Clevers, *Cell*, 2016). Our claim on the production of endogenous morphogens is not speculation, it is supported by the time-dependent production of morphogens and the presence of their respective receptors, which is measured and present in our publicly available RNA-seq datasets at GEO (GSE153185). To facilitate the reviewers and reader access to this data, we have now included this in **supplementary figure 1b**. This figure contains two additional heatmaps with the expression levels of 31 morphogens involved in heart development and their respective receptors produced by our organoids in a time-dependent fashion that have been extensively reported in the literature as important for heart development (Gessert et al, *Circulation Research*, 2010; Kwon et al, *PNAS*, 2007; Tan et al, *Fetal Diagnosis and Therapy*, 2020,; Meilhac et al, *Nature Cardiology Reviews*, 2018).

The cell type analyses all require quantification and not using the % area. Flow cytometry, single nuclei RNAS-seq and immunostaining of nuclei (eg. NKX2-5) are all options.

We thank the reviewer for this comment. We believe quantification is important to improve reproducibility and will add additional value to our data. Following the reviewer's advice, we have taken 709 new images from 138 organoids and quantified them at different heights using

double staining for nuclei and cell types of interest. The original graphs have been substituted with %number of cells in all cases.

I agree that TEMs are gold standard for t-tubules, but the presented images are not convincing. Neither are the new presented analyses using WGA staining.

We have revised the text to reduce our claim on the presence of t-tubules and just mention the possibility of t-tubules. The new text describes the presence of “tubular structures reminiscent of t-tubules” and t-tubule-like structures. In our opinion, readers can judge by themselves if these structures are sufficiently mature to be considered t-tubules or not.

The increased number of HAND1/HAND2 expression images prove spatiotemporal progress and repeated occurrence, nevertheless the authors haven't addressed the ventricular/atrial spatial compartmentalization reproducibility. Would the sectioning of those two areas always be this distinctive within the hHO and in the same area?

We have included more immunofluorescence images from more organoids showing the reproducibility and organization of ventricular/atrial spatial compartmentalization in **figure 4**.

Regarding electrophysiology in Figure 6d, the explanation given by the authors clearly states the limitations of the system used and prevents the authors from obtaining good quality quantifiable data. Therefore, it is impossible to determine whether there is a difference between the organoids under diabetic conditions or whether they don't attach as well to the MEA system (or even if they are attached to a different region as the organoids are heterogeneous). All this data is therefore flawed and cannot be used.

As we discuss in the text, organoids are not attached to the MEA system, but rather resting on top of the electrode array. Since the organoids beat robustly, they slightly bounce off the electrode with each beat, giving rise to small variations in signal amplitude. However, this does not affect the value of the data since we are not interested in signal amplitude (this parameter is not used as a measure for this reason) and all other electrophysiological parameters are wholly unaffected (frequency, QRS complex, QT interval, P and T waves). As a matter of fact, the same phenomenon occurs in a typical EKG performed by a cardiologist in a clinical setting, where slightly different amplitudes can be recorded on the electrodes on different beats due to the constant movement of the patient's heart. Of all the parameters measured, frequency and wave shape are the most valuable in MEA and EKG measurements as they allow for the identification of a number of heart phenotypes, including arrhythmia, long QT, TDP, etc. Furthermore, our electrophysiological recordings show very standard results which are to be expected (a comparison to any published work with MEA analysis will show very similar data for healthy cells). All in all, we believe this data is as solid as electrophysiology measurements get, and the method has been reported and validated before extensively too (Khan et al, 2019, *Microsystems and Nanoengineering*; Jia et al, 2018, *Journal of Neural Engineering*). Nonetheless, we have increased the value of our electrophysiology data with an alternative method, and have used live calcium imaging to measure calcium waves in live organoids. These data are now included **in Figure 6**, and fully supports the validity of the electrophysiology method. Additionally, we have included extra electrophysiological traces from other organoids to illustrate the reproducibility of the method.

To prove a point of presence of interconnecting chambers and vessels and through that a lack of hypoxic regions, proper O2 diffusion should be determined.

We thank the reviewer for this comment and advice. The presence of interconnected chambers is clearly visible in our OCT videos and figures. The presence of vascular plexus is also evident from PECAM1 staining and confocal 3D reconstruction. We do not know the levels of oxygen in our organoids and we do not make any claims in this respect, we can only assure that the chambers are not formed by cell death (as demonstrated by our FlipGFP organoids). The recent Hofbauer et al report also shows chamber formation without cell death, further reinforcing our data.

The possibility of O₂/hypoxia in larger PGDHOs or other possible explanations have not been adequately addressed.

Although hypoxia and oxygen availability are certainly interesting topics to explore, they are currently outside of the scope of this work.

I thank the authors for including the new methods for the number of cells for Seahorse assays. However, I appear to have overlooked that the cells were dissociated, rather than organoids used for the assays. This is therefore flawed as the cells will behave very differently in 2D vs 3D and there may also be substantial skewing of the data due to the dissociation/survival/attachment of the cells. Therefore, I'm not sure this current data accurately reflects the changes in metabolism caused by "diabetes" in the organoids.

We thank the reviewer for this comment, but we believe couple of clarifications are needed. The reason they cannot be loaded in the Seahorse intact is that the instrument wells are too small for these very large constructs. However other authors have described alternatives methods to bypass this limitation, such as gentle dissociation (Li et al., *Communications Biology*, 2020). In our case, the organoids were very gently dissociated with viabilities upwards of 90%, cells counted in a hemocytometer and immediately loaded in the Seahorse and measured. There is no 2D vs 3D culture, or cell attachment, as the cells are assayed on the spot and not cultured for any period of time. Both control and diabetic organoids are treated in the same exact way too, so any differences present between them are due to the disease conditions and not the technical procedure (which I insist, is the same). This method, and others like it have been reported before for organoid analysis due to the problem of organoid size (e.g. Li et al., *Communications Biology*, 2020), thus we are confident that the results are accurate and representative. We have updated the methods section in our manuscript to clarify this topic.

The numbering of figures and supplementary figures doesn't match the text in couple places and the numbering of Supplementary Figures is not in order either, making it a bit confusing to refer to.

We have reviewed all the figure numbering to ensure all figures are in order and correctly labeled.

Reviewers' Comments:

Reviewer #2:

Remarks to the Author:

The authors have somewhat addressed my concerns. The authors have overstated their novelties in the rebuttal letter (which is okay because most of this is not in the manuscript). I think the manuscript may be of interest now that most technical issues have been addressed, and perhaps readers can decide on importance.

I ask the authors to please remove "first time" phrases (eg. lines 428-429). Fetal hyperglycemia has been modelled in hPSC-CM (PMID: 29231167) and this detracts from the importance of this previous work.

The authors may also wish to revise "with well-defined QRS complexes, T and P waves", as the relevance of P waves is not immediately apparent in organoids. This is reflective of the heart, and organoids would require 2 distinct chambers where one would need much larger than the other (with a delay through some sort of node) to have bona fide P then QRS-T waves.

REBUTTAL LETTER

We want to thank the reviewers for their time and constructive criticism. A detailed point by point rebuttal follows.

Note: Original reviewers' comments have been highlighted in blue for clarity.

Reviewer #2 comments

Major concerns:

The authors have somewhat addressed my concerns. The authors have overstated their novelties in the rebuttal letter (which is okay because most of this is not in the manuscript). I think the manuscript may be of interest now that most technical issues have been addressed, and perhaps readers can decide on importance.

We thank the reviewer for this comment and are glad that they are satisfied with the changes we have made to improve the manuscript.

I ask the authors to please remove "first time" phrases (eg. lines 428-429). Fetal hyperglycemia has been modelled in hPSC-CM (PMID: 29231167) and this detracts from the importance of this previous work.

We thank the reviewer for this suggestion, and have removed all "first time" and "novel" phrases from the manuscript.

The authors may also wish to revise "with well-defined QRS complexes, T and P waves", as the relevance of P waves is not immediately apparent in organoids. This is reflective of the heart, and organoids would require 2 distinct chambers where one would need much larger than the other (with a delay through some sort of node) to have bona fide P then QRS-T waves.

We thank the reviewer for this suggestion. We have revised this sentence to better represent the data observed.